# Prediction of SARS-CoV-2-positivity from million-scale complete blood counts using machine learning

Gianlucca Zuin [1,2✉], Daniella Araujo[1,3], Vinicius Ribeiro[3], Maria Gabriella Seiler[2], Wesley Heleno Prieto[4], Maria Carolina Pintão [4], Carolina dos Santos Lazari [4], Celso Francisco Hernandes Granato[4] & Adriano Veloso[1]

## Abstract

**Background** The Complete Blood Count (CBC) is a commonly used low-cost test that measures white blood cells, red blood cells, and platelets in a person's blood. It is a useful tool to support medical decisions, as intrinsic variations of each analyte bring relevant insights regarding potential diseases. In this study, we aimed at developing machine learning models for COVID-19 diagnosis through CBCs, unlocking the predictive power of non-linear relationships between multiple blood analytes.

**Methods** We collected 809,254 CBCs and 1,088,385 RT-PCR tests for SARS-Cov-2, of which 21% (234,466) were positive, from 900,220 unique individuals. To properly screen COVID-19, we also collected 120,807 CBCs of 16,940 individuals who tested positive for other respiratory viruses. We proposed an ensemble procedure that combines machine learning models for different respiratory infections and analyzed the results in both the first and second waves of COVID-19 cases in Brazil.

**Results** We obtain a high-performance AUROC of 90 + % for validations in both scenarios. We show that models built solely of SARS-Cov-2 data are biased, performing poorly in the presence of infections due to other RNA respiratory viruses.

**Conclusions** We demonstrate the potential of a novel machine learning approach for COVID-19 diagnosis based on a CBC and show that aggregating information about other respiratory diseases was essential to guarantee robustness in the results. Given its versatile nature, low cost, and speed, we believe that our tool can be particularly useful in a variety of scenarios—both during the pandemic and after.

## Plain Language Summary

The complete blood count (CBC) is a medical laboratory test that provides information about cells in a person's blood and is extensively used to support medical decisions. This study explored the ability of a computer-based approach to automatically identify active COVID-19 infections by using CBC exams. We collected a large dataset with over one million CBC exams and the matching tests currently used to detect SARS-Cov-2 or other respiratory viruses. Our results demonstrate both the potential of this approach for diagnosing SARS-Cov-2 infection by using only CBC data, and also that considering information about other respiratory diseases in the methodology is essential to guarantee that results can be trusted. This automated computational approach can be useful in a variety of contexts during the COVID-19 pandemic and after since it is fast, low-cost, and versatile.

[1] Universidade Federal de Minas Gerais, CS Dept., Belo Horizonte, Brazil. [2] Kunumi, Belo Horizonte, Brazil. [3] Huna, São Paulo, Brazil. [4] Grupo Fleury, São Paulo, Brazil. ✉email: gzuin@dcc.ufmg.br

At the end of 2019, the Severe Acute Respiratory Syndrome Coronavirus 2 (SARS-Cov-2) appeared in the city of Wuhan, China[1], which led to a global outbreak weeks later[2]. This highly transmissible novel Coronavirus disease was named Coronavirus disease 2019 (COVID-19)[3]. At the time this article is being written, over 400 million cases of COVID-19 infections and over 5.7 million deaths have already been reported worldwide. One of the main challenges for its diagnosis is the list of initial symptoms: fever, dry cough and/or tiredness[4], which are all common in many other respiratory diseases.

Currently, the golden-standard tests for SARS-Cov-2 direct detection include the Reverse Transcription Polymerase Chain Reaction exam (or simply, RT-PCR) and the serology count analysis. The first action of the RT-PCR exam is the use of the enzyme reverse transcriptase to transform the RNA of the virus into complementary DNA. RNA is produced from a DNA molecule and presents information with which it is possible to coordinate the production of proteins. With a complementary probe to a particular virus, it is possible to verify whether the molecular content corresponds to that of the suspected infectious agent. However, in particular, for the case of SARS-Cov-2, the RT-PCR is more efficient at the peak of the infectious cycle[5]. This leads to high false-negative occurrences with a sensitivity rate of between 50% and 62% according to[6,7]. Authors in ref. [8] verified instances of over 20% infected individuals with a positive RT-PCR result only after two consecutive false-negative results. Serology exams have been found to reach a sensitivity and specificity rate of 0.95 + but only after 15–28 days of symptom onset[9]. Furthermore, both exams are relatively expensive and results take longer to process when compared with other kinds of laboratory tests, such as the complete blood count.

CBCs are extensively used for general individual diagnosis[10]. As a low-cost test that measures analyte levels of the white and red series in the blood, it is a useful tool to support medical decisions, as intrinsic variations of analytes can bring relevant insights regarding potential diseases. Patients with most kinds of infectious diseases have noticeable changes in their CBC tests. However, proving that these results can be interpreted as sufficient to support a particular diagnosis is a considerably more difficult task, as changes in analyte values could be easily confounded for different diseases' patterns.

In analyzing complete blood counts of individuals with COVID-19 infection in isolation, we find some changes to be quite characteristic of the disease[11–13]. This implies that machines, which can detect patterns not easily noticeable by humans, could be employed for automatic detection and preliminary screening of the disease. Indeed, many models have been proposed for automated COVID-19 diagnosis through CBCs and omics data. We argue that the detection performance of these models is possibly biased—or overestimated—as many patterns are not unique to SARS-Cov-2. The performance of these models will likely drop significantly as the prevalence of other respiratory viruses increases. This work employs a dataset collected between 2016 and 2021 containing exams of individuals who underwent blood tests in conjunction with RT-PCR exams throughout Brazil, both for COVID-19 and for other pathologies like Influenza-A or H1N1. More specifically, our dataset includes individuals who underwent a CBC at an interval of 60 days before or after a RT-PCR test.

For 2020 and 2021 we collected laboratory data for 900,220 unique individuals, 809,254 CBCs, and 1,088,385 RT-PCR tests, of which 21% (234,466) were positive and less than 0.2% (1679) were inconclusive. This work does not investigate demographic, prognostic, or clinical data, such as ethnicity, hospitalization, or symptomatology, as these fall out of laboratory scope. We propose modeling the task as a binary classification problem and

analyzing two distinct timeframes: one considering the early pandemic stage, namely the first wave of COVID-19 cases in Brazil; and a second stage after November 2020, when the second wave of COVID-19 started, and when we saw the emergence of a new variant of concern, *P1*, which eventually led to the health system collapse in the capital state of Amazonas in late December[14,15].

One of the key highlights of our proposed approach is the analysis of other RNA respiratory viruses. We also collected 120,807 CBCs from 2016 to 2019 of 16,940 individuals who tested positive for Influenza-A, Influenza-B or H1N1, as well as other respiratory viruses, and additionally 307,978 unlabeled CBCs. In particular, these additional CBCs included exams from the 2016 H1N1 surge in São Paulo[16], during which the population developed similar hygienic habits to the ones recommended in 2020, like social distancing and the use of masks, although at a minor scale. To the best of the authors' knowledge, this is the most extensive and comprehensive COVID-19-related dataset to date.

We follow the guidelines provided by the IJMEDI checklist[17] regarding applying machine learning to medical data, allowing for both higher quality work and an easier reproducibility and understanding of results. Our analysis focused on patients older than 18 years. We believe more experiments are necessary to assert performance for children and teens under 18 years old, but data regarding these age groups was also present in all training and test sets.

Throughout our experiments, we train an ensemble of machine learning models on this million-scale dataset to predict Sars-CoV-2-positivity. To guarantee the correct labeling of training instances, we focus on the CBC results as close to the first positive result as possible. Our analysis shows that the additional data from other RNA respiratory viruses is a fundamental aspect for properly screening COVID-19. In the absence of such information, models are prone to confound SARS-Cov-2 with other respiratory viruses or infections. This finding corroborates with many studies that raised concerns regarding bias in COVID-19 research[18–20]. We also demonstrate the necessity of maintaining a model as up-to-date as possible to allow any machine learning model to keep up with the different stages of a pandemic surge. Our model retains high-performance values across multiple evaluation scenarios and on simulations with varying prevalences of COVID-19, properly differentiating Sars-CoV-2 from other confounding viruses, thus demonstrating the robustness of our approach.

## Methods

**Data**. The Fleury database structure was created on 10/1997 using an InterSystems Caché and Ensemble, version 1.4 (Caché, InterSystem, 2018; https://docs.intersystems.com/; November 2020), a high-performance architecture that is commonly used to develop software applications for healthcare management (Cambridge MA). The database was built using standard healthcare industry practices to ensure accuracy, completeness, and security of data collected. The results of the laboratory tests are automatically inserted into a Microsoft SQL database after verification of the RT-PCR output. Within a few seconds, data are replicated to the Cache Database—Intersytems—for permanent storage. Once stored in the database, the result is made available for patients. All users have a username and password, maintained by AD Windows (Active Directory). All registry changes to the database are tracked through a log and are restricted to users with high-level administrative permissions. Information is kept secure through a separate network firewall, accessed only by authorized persons within the Fleury Group's domains. Data stored in this database has been used previously in several clinical studies before theCOVID-19 outbreak[21–26].

This project was submitted, evaluated, and approved by the Research Ethics Committee (CEP) of Grupo Fleury (CAAE: 33790820.3. 0000.5474), duly qualified by the National Research Ethics Committee (CONEP) of the National Health Council of Brazil. The Research Ethics Council (CEP) is an interdisciplinary and independent collegiate of public relevance, consultative, deliberative, and of educational character, created to defend the interests of research participants in their integrity and dignity as well as to contribute research development within highest ethical standards. By decision of the CEP, since this project uses retrospective and anonymized data, there is no need to apply an e Free and Informed Consent Term (TCLE) to participating patients.

The CBC measurements were obtained from EDTA-K3 collected peripheral blood samples analyzed by the Automated Hematology Analyzer XT or XN series from Sysmex (Sysmex Corporation, Kobe, Japan). In total, 72 pieces of equipment are distributed in 36 laboratories over the country. Red blood cells (RBC) and platelets were counted and sized by direct current impedance with hydrodynamic focusing and heath flow direct current (DC) detection was used. The hematocrit was determined from the RBC pulse height. The hemoglobin was measured using sodium lauryl sulfate spectrophotometry. CBCs also include the physical features of the RBC: Mean corpuscular volume (MCV) is a measurement of the average size of red blood cells; Mean corpuscular hemoglobin (MCH) is a calculated measurement of the average amount of hemoglobin; Mean corpuscular hemoglobin concentration (MCHC) is a calculated measurement of the average concentration of hemoglobin; and Red cell distribution width (RDW) is a measurement of the variation in RBC size. The white blood cells (WBC) and six-part differential were determined by fluorescence flow cytometry. Specifically, the WBC subpopulations were separated based on cell complexity (side-scattered fluorescent intensity), cell size (forward scattered light), and fluorescence signal (side fluorescent light).

Quality control is performed daily using three control levels (high, normal, and low) for each parameter. Measurements are analyzed using the InsightTM Interlaboratory Quality Assessment Program for Sysmex hematology analyzers, where data from users worldwide are compared. To guarantee equivalence and reproducibility of our analysis and enable the use of common reference intervals for different measurement procedures[27], harmonization of equipment is performed in accordance with the Clinical and Laboratory Standards Institute's (CLSI) guidelines[28]. Results are accepted if the percentage difference is less than 50% of the total error for each parameter, which allows us to devise reference values for each measurement[29,30].

**Complete blood count and model features**. A complete blood count (or simply, CBC) is a common blood test used for a variety of reasons, including the detection of disorders and infections. A CBC test measures several components and features in the blood, including RBC, which carry oxygen; Hemoglobin, the oxygen-carrying protein in red blood cells; Hematocrit, the proportion of red blood cells to the fluid component; WBC, which fight infection (i.e., Monocytes, Lymphocytes, Eosinophils, Basophils, Neutrophils); and Platelets, which help with blood clotting.

Abnormal increases or decreases in cell counts may indicate an underlying biological process taking place, like inflammation or immune response. Also, values such as the Neutrophil-Lymphocyte ratio, Platelet-Monocyte ratio, or the Platelet-Lymphocyte ratio are recognized as inflammatory markers[31]. Table 1 shows analyte means and standard deviations, as well as the employed units of measure in each of our cohorts. We can easily identify some patterns that might help us in sorting

COVID-19 infected patients from the remaining ones. We can also clearly perceive that the distributions for each gender are slightly different. This is to be expected, as it is known that CBC values vary with age and gender[32]. However, introducing an explicit gender variable into our model could entail bias. To avoid this, we instead normalize each analyte by the corresponding gender and age reference values devised by Grupo Fleury, thus building a unified model that considers CBC analyte values regardless of gender.

Specifically, we perform normalization by employing the reference ranges as a pivot. Let $R$ be the reference values of an analyte, the general formula scaling features is given as

$$x' = \frac{x - \Omega(R(x|\text{sex} = s, \text{age} = a))}{O(R(x|\text{sex} = s, \text{age} = a)) - \Omega(R(x|\text{sex} = s, \text{age} = a))} \quad (1)$$

where $x$ is an original value, $x'$ is the normalized value, $R(x|\text{sex} = s, \text{age} = a)$ describes the reference values for $x$ given the sex $s$ and age $a$ of a patient, and $\Omega$ and $O$ represent the lower and upper bounds respectively. For example, supposing a male adult presents a 5.0 millions/mm$^3$ RBC and knowing that the reference values lie in the range $[4.30 - 5.70]$, we first subtract 4.30 from 5.0 and divide the result by 1.4 (the difference between the maximum and minimum reference values), thus obtaining the normalized 0.5 RBC count. Consequently, normalized values above 1 represent abnormally high cell counts. Likewise, normalized values below 0 represent abnormally low counts. Our model analyzes normalized cell counts and their corresponding pairwise ratios as potential features for building our models.

The performance of machine learning methods are heavily dependent on the choice of features on which they are applied[33]. For this reason, much of the current effort in deploying such algorithms goes into the design of preprocessing pipelines and data transformations that result in a representation of data that can support effective machine learning[33–35]. The process of using available features to create additional ones to improve model performance is often called 'feature engineering', a predominantly human-intensive and time-consuming step that is central to the data science workflow. It is a complex exercise, performed in an iterative manner with trial and error, and mostly driven by domain knowledge[36]. Recently, many studies have shown the benefits of automatizing this process by creating candidate features in a domain-independent and data-driven manner followed by an effective method of feature selection. This way it is possible not only to improve model correctness but also to discover powerful new features and processes that could be additional candidates for domain-specific studies[36–38]. We avoid potential spurious correlations by confirming that all selected features present a strictly non-zero impact on model output after n-fold cross-validation.

**Inclusion–exclusion criteria**. The scale of our dataset allows us to produce high-quality training sets and massive validation sets. Table 2 provides the gender and RT-PCR results distribution employed for training and evaluating our models. In addition to SARS-Cov-2, Influenza-A, Influenza-B, and Influenza-H1N1, our dataset also comprehends a variety of other viruses, including Coronavirus OC43, Human Metapneumovirus A, Adenovirus, Parainfluenza 1, Coronavirus HKU1, Enterovirus B, Parainfluenza 2, Coronavirus NL63, Respiratory Syncytial Virus A, Mycoplasma pneumoniae, Respiratory Syncytial Virus B, Rhinovirus, Human Metapneumovirus B, Coronavirus 229E, Chlamydophila pneumoniae, Bordetella pertussis, Parainfluenza 3, Bocavirus, and Parainfluenza 4. We argue that taking this variety of confounding viruses into consideration is of utmost importance to learn models that are specific for COVID-19.

**Table 1 Mean and standard deviation for all considered cell counts in each cohort. N = 1,138,728 CBCs.**

| Analyte | Covid-19 (+) | Covid-19 (−) | Influenza (+) | Other Viruses (+) | Entire Data |
|---|---|---|---|---|---|
| *Male patients* | | | | | |
| RBC ($10^{12}$/L) | 5.06 ± 0.52 | 4.21 ± 0.98 | 4.73 ± 0.60 | 3.67 ± 0.87 | 4.28 ± 0.96 |
| Hemoglobin (g/dl) | 14.9 ± 1.4 | 12.4 ± 2.8 | 14.0 ± 1.7 | 10.8 ± 2.5 | 12.6 ± 2.7 |
| Hematocrit (%) | 43.8 ± 4.0 | 36.8 ± 7.9 | 41.0 ± 4.9 | 31.7 ± 7.3 | 37.4 ± 7.7 |
| MCV (fL) | 86.8 ± 4.7 | 88.1 ± 6.4 | 87.0 ± 6.7 | 86.9 ± 8.0 | 88.0 ± 6.2 |
| MCH (pg/cell) | 29.5 ± 1.9 | 29.6 ± 2.3 | 29.6 ± 2.3 | 29.6 ± 2.6 | 29.5 ± 2.2 |
| MCHC (g/dL) | 34.1 ± 1.1 | 33.6 ± 1.4 | 34.0 ± 1.1 | 34.1 ± 1.4 | 33.6 ± 1.4 |
| RDW (%) | 13.0 ± 1.0 | 14.3 ± 2.2 | 13.6 ± 1.2 | 15.1 ± 2.1 | 14.1 ± 2.2 |
| WBC ($10^9$/L) | 6.07 ± 2.37 | 8.07 ± 3.81 | 6.96 ± 2.81 | 5.87 ± 4.69 | 8.02 ± 3.81 |
| Monocytes ($10^9$L) | 0.66 ± 0.29 | 0.68 ± 0.35 | 0.75 ± 0.37 | 0.66 ± 0.46 | 0.66 ± 0.34 |
| Lymphocytes ($10^9$L) | 1.40 ± 0.72 | 1.67 ± 1.05 | 1.23 ± 0.92 | 1.25 ± 1.40 | 1.54 ± 0.99 |
| Eosinophils ($10^9$/L) | 0.07 ± 0.09 | 0.18 ± 0.20 | 0.07 ± 0.10 | 0.10 ± 0.16 | 0.15 ± 0.20 |
| Basophils ($10^9$/L) | 0.02 ± 0.02 | 0.03 ± 0.02 | 0.02 ± 0.01 | 0.02 ± 0.02 | 0.03 ± 0.02 |
| Neutrophils ($10^9$/L) | 3.92 ± 2.22 | 5.53 ± 3.50 | 4.90 ± 2.57 | 4.08 ± 3.93 | 5.64 ± 3.57 |
| Platelets ($10^9$/L) | 195.7 ± 56.7 | 222.0 ± 102.3 | 182.9 ± 63.6 | 145.8 ± 115.6 | 222.7 ± 99.9 |
| *Female patients* | | | | | |
| RBC ($10^{12}$/L) | 4.57 ± 0.44 | 4.03 ± 0.75 | 4.62 ± 0.67 | 3.75 ± 0.78 | 4.05 ± 0.75 |
| Hemoglobin (g/dl) | 13.3 ± 1.2 | 11.8 ± 2.1 | 13.6 ± 1.9 | 11.0 ± 2.1 | 11.8 ± 2.1 |
| Hematocrit (%) | 39.8 ± 3.4 | 35.4 ± 6.1 | 40.3 ± 5.4 | 32.8 ± 6.4 | 35.6 ± 6.0 |
| MCV (fL) | 87.3 ± 5.0 | 88.3 ± 6.3 | 87.7 ± 6.7 | 87.9 ± 8.1 | 88.3 ± 6.2 |
| MCH (pg/cell) | 29.2 ± 2.0 | 29.3 ± 2.3 | 29.7 ± 2.3 | 29.4 ± 2.7 | 29.3 ± 2.2 |
| MCHC (g/dL) | 33.5 ± 1.0 | 33.2 ± 1.3 | 33.8 ± 1.2 | 33.5 ± 1.4 | 33.2 ± 1.3 |
| RDW (%) | 13.1 ± 1.1 | 14.2 ± 2.1 | 13.7 ± 1.3 | 14.9 ± 2.1 | 14.1 ± 2.1 |
| WBC ($10^9$/L) | 5.87 ± 2.40 | 8.03 ± 3.71 | 7.11 ± 3.15 | 6.62 ± 4.63 | 7.84 ± 3.66 |
| Monocytes ($10^9$/L) | 0.56 ± 0.24 | 0.62 ± 0.32 | 0.70 ± 0.35 | 0.61 ± 0.43 | 0.60 ± 0.31 |
| Lymphocytes ($10^9$/L) | 1.54 ± 0.80 | 1.85 ± 1.05 | 1.36 ± 0.95 | 1.54 ± 1.40 | 1.78 ± 1.02 |
| Eosinophils ($10^9$/L) | 0.06 ± 0.08 | 0.16 ± 0.18 | 0.075 ± 0.11 | 0.09 ± 0.18 | 0.15 ± 0.18 |
| Basophils ($10^9$/L) | 0.02 ± 0.01 | 0.03 ± 0.02 | 0.01 ± 0.01 | 0.02 ± 0.02 | 0.03 ± 0.02 |
| Neutrophils ($10^9$/L) | 3.68 ± 2151.51 | 5.39 ± 3.36 | 4.94 ± 2.97 | 4.56 ± 3.81 | 5.29 ± 3.34 |
| Platelets ($10^9$/L) | 222.6 ± 63.0 | 249.2 ± 101.4 | 185.0 ± 69.1 | 188.9 ± 123.8 | 248.4 ± 100.4 |

*Safe labeling.* It is worth mentioning that CBCs and RT-PCRs are part of different exam batteries, and are therefore often collected on different dates for the same individual. Thus, an important decision is the ideal time frame between the collection of a CBC and that of the RT-PCR test used to validate its label. It is challenging to validate the precise moment the infection has initiated considering the lack of information concerning the onset of symptoms. We also observed abnormalities in the CBCs associated with recovered individuals. These differences could be related to drug usage and/or other therapies, or be due to symptoms that persist even after the virus has been eliminated. In this context, we have the hypothesis that CBCs, even when associated with a positive RT-PCR, may be affected by treatment-related effects. Figure 1 shows the concentration distribution of some analytes along with the disease progression time frame. The lower the ratio between white blood cells (WBC) and red blood cells (RBC), the higher is the probability of the individual being positive for COVID-19. Additionally, we observed that the lowest value for this ratio lies on day 0. Since our working dataset consists of patients who went to one of Grupo Fleury's laboratories to undertake an exam, we hypothesize that the search for an RT-PCR, in particular for patients who obtained a confirmatory diagnosis of COVID-19, might be associated with the start of symptoms onset, explaining this particular pattern. We did not observe similar behavior for other evaluated viruses, perhaps due to the relative difference in public awareness/concern regarding SARS-Cov-2 and Influenza infections.

Furthermore, we also observed that most analytes tend to present abnormal values for up to 30 days. This might be related to the natural evolution of COVID-19 onto the inflammatory stage, the effects of treatments, or even long-lasting effects on patients' immunological systems. We concluded that the safest and most effective gap to use for labeling CBCs with RT-PCRs outcomes' is the 24-h window centered on the first positive RT-PCR result of an individual, with the remaining frames being highly uncertain about a positive diagnosis, and thus discarded.

*Removing gender and age biases.* Supplementary Figure 1 presents the age distribution of each pathology subset. We verify a small prevalence in male positive COVID-19 cases and Female positive Influenza. To address this, we sub-sampled the training sets to remove possible biases that could jeopardize learning and validated unsampled data to properly verify model behavior in real-world scenarios.

*Removing possible false-negative cases.* Another point of attention is the possible existence of false-negative results for RT-PCR exams. In particular, we often see cases of the same individual having negative results interspersed with two or more positive results. Therefore, it is also necessary to carry out a preprocessing step to guarantee authenticity of negative labels and to ensure that the model is as faithful as possible to the real scenario of COVID-19, and not to the limitations of the RT-PCR exam. We filter out any negative RT-PCR results issued after the first positive RT-PCR result, thus focusing our analysis on pre-covid individuals and those on the preliminary stages of the infection. We also consider individuals that never had any contact with SARS-Cov-2 in our negative cohort, namely individuals with exams dating before 2020.

**Outbreak waves.** Table 2 also shows training and validation sets for the two waves that occurred during the COVID-19 outbreak in Brazil. The training set for the first wave comprises labeled CBCs acquired until 26 June 2020, whilst its validation set

**Table 2 Entire dataset, training sets, and validation sets for the two waves that occurred during the Brazilian COVID-19 outbreak.**

| | CBC (+) | CBC (−) | | | | |
|---|---|---|---|---|---|---|
| Gender | COVID-19 (+) | COVID-19 (−) | Influenza-A (+) | Influenza-B (+) | Influenza-H1N1 (+) | Other viruses (+) |
| *Entire data* | | | | | | |
| Male | 11.3% (122,793) | 34.0% (369,787) | 46.7% (3160) | 46.5% (1384) | 48.4% (4108) | 59.5% (20,107) |
| Female | 10.3% (111,673) | 44.4% (482,453) | 53.3% (3604) | 53.5% (1588) | 51.6% (4380) | 40.5% (13,691) |
| *Training set: first wave data* | | | | | | |
| Male | 12.9% (5859) | 9.8% (4469) | 4.2% (1895) | 2.1% (975) | 6.0% (2742) | 12.8% (5825) |
| Female | 12.1% (5527) | 15.2% (6918) | 4.9% (2223) | 2.8% (1214) | 6.9% (3118) | 10.3% (4656) |
| *Validation set: first wave data* | | | | | | |
| Male | 4.9% (5808) | 37.6% (44,637) | 1.0% (1113) | <0.1% (188) | 1.4% (1660) | 2.3% (2710) |
| Female | 4.7% (5647) | 43.4% (51,550) | 1.1% (1343) | <0.1% (134) | 1.6% (1842) | 1.7% (2028) |
| *Training set: second wave data* | | | | | | |
| Male | 25.9% (24,104) | 10.5% (9770) | 2.0% (1895) | 1.0% (975) | 3.1% (2742) | 6.3% (5825) |
| Female | 24.2% (22,404) | 15.1% (14,088) | 2.3% (2223) | 1.3% (1214) | 3.3% (3118) | 5.0% (4656) |
| *Validation set: second wave data* | | | | | | |
| Male | 4.5% (11,860) | 38.9% (101,655) | 0.4% (1113) | <0.1% (188) | 0.6% (1660) | 1.0% (2710) |
| Female | 4.3% (11,021) | 48.1% (125,776) | 0.5% (1343) | <0.1% (134) | 0.7% (1842) | 0.8% (2028) |

Training sets were obtained after applying the inclusion–exclusion criteria to the entire data and downsampling the COVID-19(-) class in the training sets to account for class unbalance. We considered October 1st as the split point between the first and second wave data to eliminate possible incubation periods before the start of the second wave in early November. As such, validation for the first wave encompasses data from late June to late September, and validation for the second wave ranges from early October to late February. N = 1,138,728 CBCs.

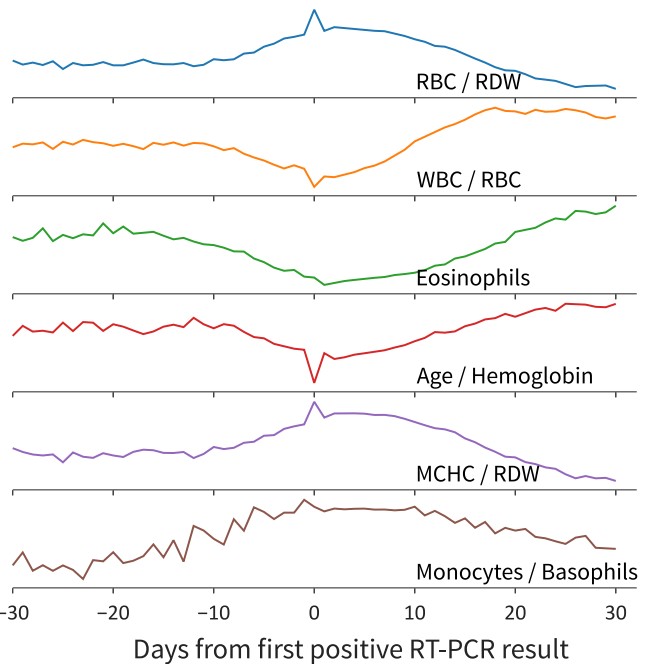

**Fig. 1 Analytes average progression through COVID-19 disease course.** Average values of the most impactful analytes along with the disease time frame, from 30 days before the first positive RT-PCR result up to 30 days after. N = 120,726 patients.

comprises labeled CBCs acquired between 27 June 2020 and 05 September 2020. The training set for the second wave comprises labeled CBCs acquired until 30 September 2020, whilst its validation set comprises labeled CBCs acquired between 01 October 2020 and 28 February 2021. Both training and validation sets contain data corresponding to viruses other than SARS-Cov-2: the training sets contain instances from 2016 to 2018, while the validation sets contain instances from 2019.

**Statistics and reproducibility.** Our main objective is to demonstrate that training a model directly on COVID-19 data is not enough to guarantee robustness if multiple respiratory infections are present, as might be expected to occur in a possible COVID-19 endemic scenario. This is true even in the case of a massive dataset such as the one we employed for our study. Thus, we built resilient models for a pre-selected case of core confounding viruses and showed that we can retain similar COVID-19 detection performance in a scenario containing the prevalence of COVID-19 as well as achieving high discriminatory figures in low-prevalence scenarios with an abundance of other respiratory infections. Furthermore, we also demonstrate that the model indeed learns useful relationships between CBC patterns and other respiratory infections. To ensure the relevance of the results, we assess the statistical significance of our measurements through a pairwise *t*-test[39] with *p*-value ≤ 0.05 and through 5-fold cross-validation.

*Model training.* Our models were trained with the objective of distinguishing CBCs (+) from CBCs (−) (refer to Table 2). We followed a stacking procedure, that is, the training stage consists of creating multiple specialized models for each of the viruses

considered (i.e., COVID-19, Influenza-A, Influenza-B, Influenza-H1N1, and other viruses) and then combining their outputs to obtain a final prediction about the target disease. We divided the training samples into two equally sized batches. The first one was used to train the specialized models and the second one to train the final stacked model. Each specialized model only had access to label information regarding the corresponding virus, and the stacking model employs CBC (+) and CBC (−) labels.

Both specialized models as well as the final stacking model were trained with lightGBM[40], a fast implementation of a tree-based gradient boosting technique. We employed the SHAP algorithm[41–43] to obtain an interpretation of the model's prediction, allowing us not only to have a probability that a specific CBC is associated with a positive RT-PCR for COVID-19 but also an explanation consisting of the feature importance leading to the model decision. We assessed performance by calculating AUROC, sensitivity and specificity in the validation sets as well as running 5-fold cross-validation in the training sets. Supplementary Figure 2 illustrates the proposed approach's pipeline. We performed extensive grid-search for hyperparameter tuning for all the aforementioned models. Our final models employ 100 Gradient-Boosted Decision Trees estimators with a maximum tree depth of 50 and a maximum number of leaves of 50. The learning rate was set to $2e^{-1}$ optimizing the binary cross-entropy function.

*Selecting specialized models.* Not all CBC analytes are relevant features for differentiating the base targets (i.e., each virus), and some features may be detrimental to the task. To find a set of relevant features, we represent the model space as a directed acyclic graph (DAG) in which each node represents a distinct feature subset, and vertex $A \rightarrow B$ is connected if $B$ can be reached by simple feature addition from $A$, thus representing a transitive reduction of the more complex combinatorial complete model space. This modeling approach presents two desirable properties: the first being that any vertex is reachable from the [Ø] model, the second being that, for any feature set path, there exists a topological ordering, an ordering of all vertices into a sequence such that for every edge, the start vertex occurs earlier in the sequence than the ending vertex of the edge. These properties imply a partial ordering of the graph starting from the root node, which allows us to search it in an orderly manner. We apply the A* algorithm[44], employing as heuristic the AUROC of the model represented by the feature set of a given vertex. We hypothesize that there exists a set of optimal feature expansions that lead to the best-performing models for each specific base task. This allows us to search the $N!$ combinatorial space of feature subsets to select the best performing specialized models.

*Learning the final model.* Our stacking definition extends all previously related COVID-19 learning approaches by building specialized models targeted at confounding viruses. When building the final model, we can expect to learn prediction relationships between COVID-19 and other respiratory infections. For example, in a scenario of a moderately high chance of Influenza, we would need an exceedingly high COVID-19 probability to confirm a positive infection hypothesis.

**Reporting summary**. Further information on research design is available in the Nature Research Reporting Summary linked to this article.

## Results

**Features and model effectiveness for COVID-19 identification.** Our first set of experiments is dedicated to validating that CBCs are useful sources of information for identifying SARS-Cov-2

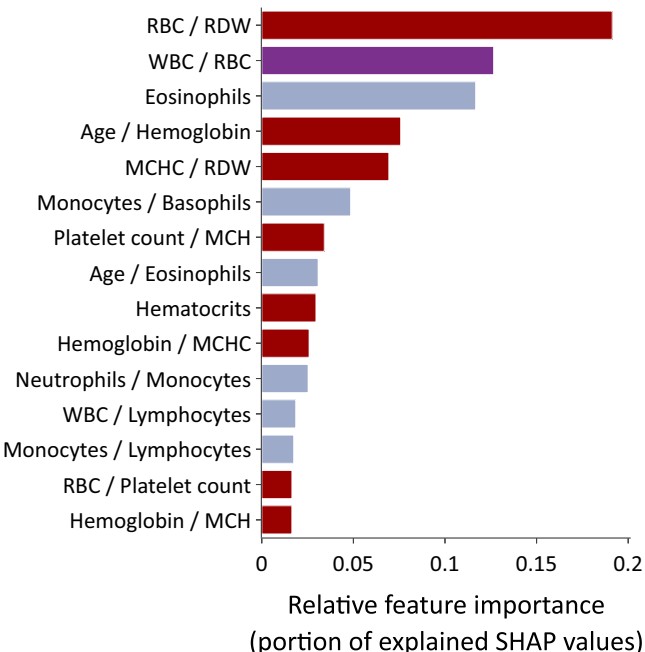

**Fig. 2 Analyte feature importance of the COVID-19 specialized model.** Features in red are exclusively from the red series (roughly 60% of total importance). Features in gray are exclusively from the white series (roughly 26% of total importance). Features in purple involve analytes from both red and white series (roughly 14% of total importance). $N = 103{,}822$ CBCs.

virus infection. It is worth mentioning that in this initial experiment we did not employ information about infections other than COVID-19 while training the model, that is, CBC (−) is composed only by the sub-population in COVID-19 (−). We trained a COVID-19 model with the labeled CBCs within the first two quarters of 2020 and evaluated it onto the labeled CBCs within the third quarter of 2020. Supplementary Fig. 3 shows the AUROC improvement as we proceed to include more features in the COVID-19 model. We can verify that employing only three features is already enough to surpass the 0.85 AUROC mark. Our final COVID-19 model achieves an AUROC of 0.922, specificity of 0.918, and sensitivity 0.824, thus clearly indicating the potential of employing large volumes of CBCs to identify SARS-Cov-2 virus infection. Figure 2 presents the 15 most important features identified by our algorithm as well as their contribution to the final specialized COVID-19 model prediction.

**SARS-Cov-2 Mutations and Variants.** By mid-November 2020, Brazil entered the second wave of COVID-19, which eventually led to the collapse of the health system in Manaus, capital of Amazonas, a state in Brazil[45]. One of the explanations raised by the local government was the emergence of a new COVID-19 variant, known as 20J/501Y.V3—or simply P.1[14]. To evaluate the performance of our COVID-19 model as the SARS-Cov-2 virus mutates, we trained it at two distinct points in time. The first one, which we will refer to as the "First-wave model", was trained using the training set associated with the first wave (as shown in Table 2). The second, which we will refer to as the "Second-wave model" was trained using the training set associated with the second wave in Brazil (as shown in Table 2).

Figure 3 presents the AUROC obtained after the application of each of these two models during the pandemic, up to March-2021, considering a 7 days sliding window, as well as the respective COVID-19 prevalence (i.e., the proportion of positive cases over all RT-PCR exams in a given period). We investigate

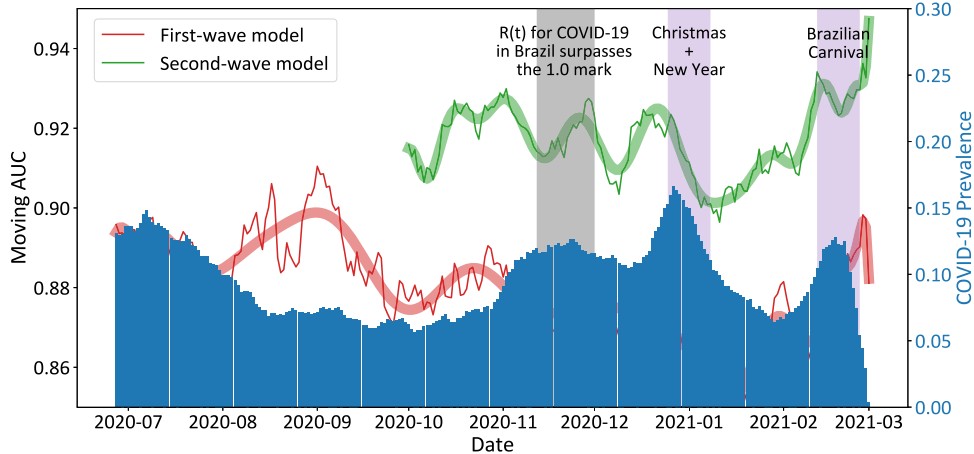

**Fig. 3 AUROC fluctuation over time considering a 7-day sliding window.** The red line represents the model trained only on the first wave of COVID-19 in Brazil data (up to 2020-06) while the green line represents a model trained with data immediately before the start of the second wave of COVID-19 in Brazil (up to 2020-10). Thinner lines depict the measured AUROC values while thicker lines illustrate their respective trends. The second wave model can retain performance during the second wave while the performance of the first-wave model deteriorates. Key events are marked in gray and purple. $N = 357,956$ CBCs.

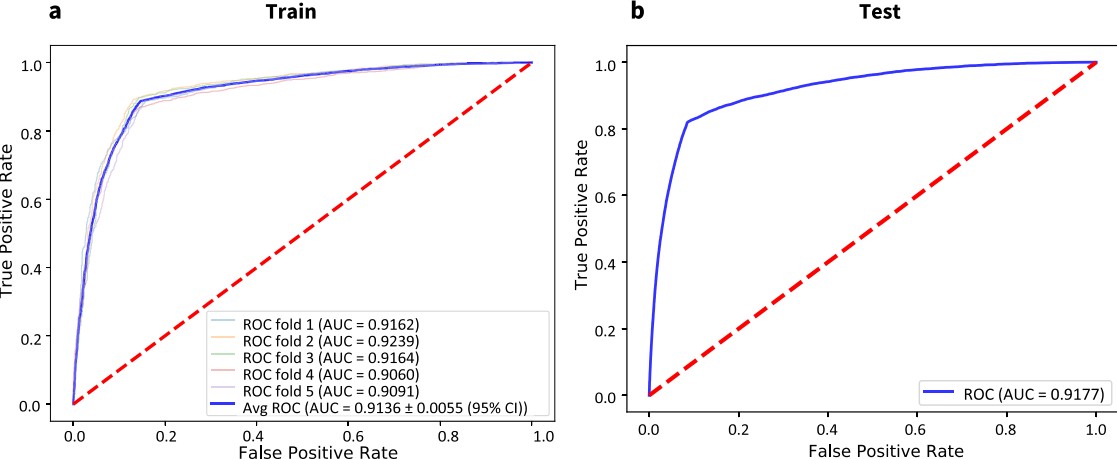

**Fig. 4 AUROC values for the proposed stacking model. a** Cross-validation performance. **b** Test set performance. $N = 91,014$ train CBCs and 261,630 test CBCs.

three periods of interest: $R(t) > 1.00$, a period in which the SARS-Cov-2 reproduction number was above 1.00 uninterruptedly for several days. During this period the virus spread quickly through the entire country; Christmas + New Year day, a period in which families reunite, spreading the virus and resulting in a clear increase in COVID-19 cases and observed in the entire country; and Carnival, a period in which large crowds fraternize. Carnival events were canceled for 2021, but many gatherings were reported in some regions of the country, such as Rio de Janeiro, Natal, and Recife.

The performance of the First-wave model seems to deteriorate with time, mostly as a result of periods of high COVID-19 prevalence due to SARS-Cov-2 variants. On the other hand, the Second-wave model reaches AUROC values as high as 0.952. Interestingly, the periods we analyzed affected the two models in different ways, but the experiment highlights the importance of retraining the models so that they can account for eventual virus variants.

**Identifying SARS-Cov-2 in the presence of other RNA respiratory viruses.** Our previous set of experiments verified the performance of our models in predicting the COVID-19 RT-PCR result from complete blood counts. However, a key concern

remained regarding the ability to distinguish between different respiratory viruses. Thus, after a careful study, we further trained specialized models in an attempt to predict the RT-PCR result for various types of Influenza and other respiratory viruses. Our approach employs stacking to combine the outputs of each specialized model (i.e., COVID-19, Influenza-A, Influenza-B, H1N1, etc.) to perform a final prediction for COVID-19. Specifically, we used half of the training data to learn specialized models, and the other half to train the final stacked model. As illustrated in Fig. 4, our stacked COVID-19 model achieves performance as high as 0.913 (cross-validation on the stacking training sets shown in Table 2) and 0.917 (using stacked training and validation sets shown in Table 2) while retaining 0.80 sensitivity and 0.91 specificity.

While the stacked model achieves high-performance predicting COVID-19, it is also important to verify its specificity by analyzing the predictions performed for individuals infected with viruses other than SARS-Cov-2. Figure 5 shows how different models perform specifically on individuals that were infected by some viruses in 2019. The ideal result would be all predictions being negative for COVID-19. As discussed before, models trained solely on SARS-Cov-2 data are very effective in

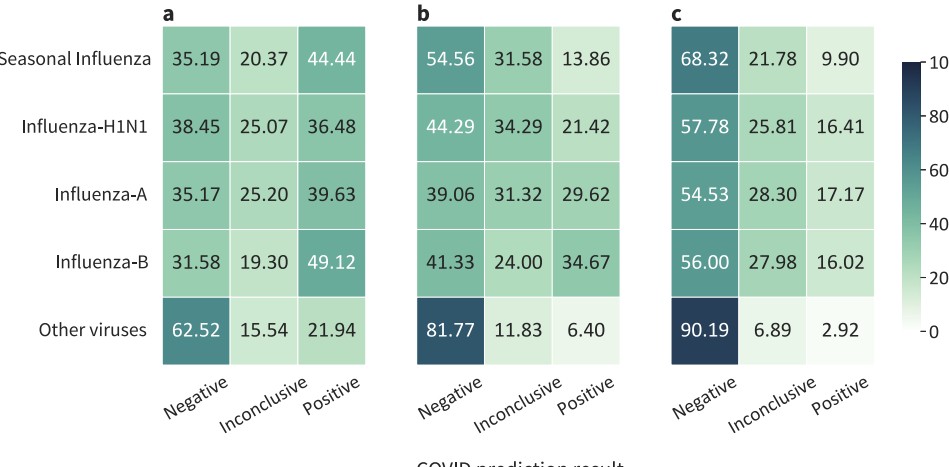

**Fig. 5 Results of different models evaluated on 2019 individuals with confirmed RT-PCR results for diverse viruses, including Influenza-A, Influenza-B, Influenza-H1N1, and Seasonal Influenza. a** Model trained only on SARS-Cov-2 data. CBC (−) includes only COVID-19 (−). **b** Model trained using data of diverse viruses, including SARS-Cov-2. CBC (−) also includes viruses other than SARS-Cov-2. **c** The stacked model. CBC (−) also includes viruses other than SARS-Cov-2. Specialized models are trained using half of the training sets, and then these specialized models are combined using the other half of the training sets. $N = 11\,116$ CBCs.

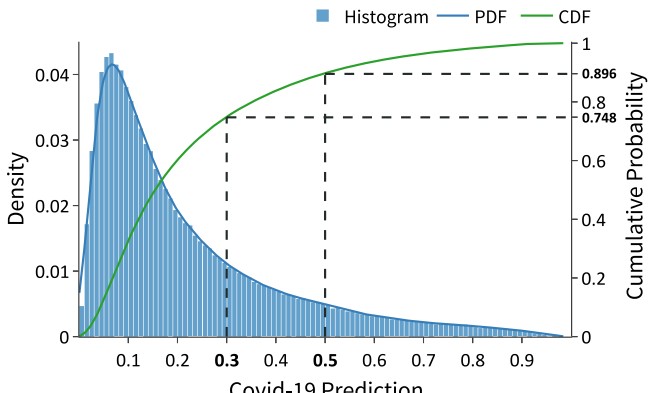

**Fig. 6 Stacked model probability of predicting COVID-19 on 2019 data.** Nearly 90% of the cases lie below the 50% covid probability threshold, with roughly 75% being concentrated below the 30% probability threshold. $N = 307{,}978$ CBCs.

identifying COVID-19 cases, but the result of 2019 data indicates that these models performed poorly on other viruses (Fig. 5a). Including viruses other than SARS-Cov-2 during training increases the performance on 2019 data (Fig. 5b). The stacked model proves to be much more specific for COVID-19 than both previous models (Fig. 5c).

Figure 6 also investigates the specificity of the stacked model by showing the prediction distribution on the 2019 data (i.e., individuals infected by a virus other than SARS-Cov-2). The stacked model associates 0−10% COVID-19 probability to roughly 44% of the predictions on 2019 data. Furthermore, the stacked model correctly places almost 80% of the evaluated individuals below the 30% COVID-19 prediction mark, with over 40% being placed below the 7% probability mark.

**Simulating endemic-pandemic scenarios**. We also considered how the model would perform in an endemic scenario in which individuals infected with SARS-Cov-2 could be scarce, and where other types of confounding viruses might be present. To simulate different scenarios, we evaluate the stacked model on data with

different COVID-19 prevalences. Specifically, we sample exams from the second wave validation and 2019 data to control the COVID-19 prevalence. The main goal is to stress the stacked model by presenting cases before any safety and/or social distancing policies could take place, in an attempt to mimic what could happen in an endemic future. These results are summarized by the AUROC, sensitivity, and specificity numbers for each evaluated COVID-19 prevalence presented in Table 3. To guarantee statistical significance, we perform 30 repetitions of each simulation and present the respective 95% confidence intervals. The stacked model proved to be robust on varying levels of COVID-19 prevalence.

## Discussion
The CBC is a simple and inexpensive exam. It is part of most laboratory routines, so *"astute practitioners may use nuances and clues from the CBC in many clinical situations"*[10]. Liu et al.[46] devised a high accuracy risk assessment tool that can predict the mortality for COVID-19 through CBCs. Li et al.[47] verified that the low count of white blood cells is related to COVID-19 severity by analyzing 12 death cases of COVID-19 and 18 individuals with moderate to severe symptoms, verifying low lymphocyte percentage in most of the cases. Although our dataset had no indicator of severity, we did find a drop in lymphocyte count the closer individuals were to their first positive RT-PCR results, corroborating this finding. Furthermore, we also verified many other analytes that shared a similar pattern. Although more research is needed, we believe that the key analytes indicated by our model might provide possibilities for future research. Literature suggests that there might be existing intrinsic relationships between analytes that might be characteristic of COVID-19. For instance, Nalbant et al.[48] found that the neutrophil/lymphocyte ratio (NLR) might be particularly typical of COVID-19 infection. However, there is a profusion of other possible promising ratios and patterns currently being under-analyzed for the sake of COVID-19 diagnosis. One of the secondary goals of this work is to investigate this hypothesis, and we confirmed that our search algorithm tends to favor ratios over analyte count values.

We identified several works attempting to exploit blood counts to detect COVID-19 with the help of machine learning algorithms. Avila et al.[49] trained a naive Bayes classifier with data from 510

**Table 3 COVID-19 endemic and pandemic simulations. AUROC, Specificity and Sensitivity, and the respective confidence intervals for different COVID-19 prevalence simulations under 95% confidence.**

| COVID-19 prevalence | AUROC | Specificity | Sensitivity |
|---|---|---|---|
| 1% | 0.928 ± 0.093 | 0.875 ± 0.018 | 0.913 ± 0.152 |
| 2% | 0.881 ± 0.117 | 0.877 ± 0.024 | 0.812 ± 0.250 |
| 3% | 0.917 ± 0.046 | 0.874 ± 0.016 | 0.873 ± 0.099 |
| 4% | 0.922 ± 0.037 | 0.882 ± 0.033 | 0.896 ± 0.087 |
| 5% | 0.918 ± 0.046 | 0.874 ± 0.012 | 0.879 ± 0.104 |
| 6% | 0.909 ± 0.041 | 0.874 ± 0.032 | 0.857 ± 0.116 |
| 7% | 0.910 ± 0.024 | 0.883 ± 0.018 | 0.840 ± 0.083 |
| 8% | 0.904 ± 0.054 | 0.879 ± 0.036 | 0.849 ± 0.102 |
| 9% | 0.907 ± 0.046 | 0.872 ± 0.025 | 0.871 ± 0.085 |
| 10% | 0.896 ± 0.059 | 0.871 ± 0.025 | 0.848 ± 0.118 |
| 20% | 0.916 ± 0.029 | 0.866 ± 0.025 | 0.878 ± 0.045 |
| 30% | 0.906 ± 0.021 | 0.871 ± 0.018 | 0.862 ± 0.059 |
| 40% | 0.911 ± 0.016 | 0.871 ± 0.024 | 0.873 ± 0.032 |
| 50% | 0.913 ± 0.032 | 0.886 ± 0.030 | 0.863 ± 0.028 |
| 60% | 0.901 ± 0.015 | 0.868 ± 0.031 | 0.852 ± 0.037 |
| 70% | 0.906 ± 0.021 | 0.867 ± 0.033 | 0.858 ± 0.035 |
| 80% | 0.902 ± 0.040 | 0.869 ± 0.074 | 0.854 ± 0.025 |
| 90% | 0.911 ± 0.030 | 0.889 ± 0.081 | 0.864 ± 0.022 |

*N* = 30 simulations with 20,000 unique patients each.

individuals admitted to hospitals presenting COVID-19-like symptoms with a reported AUROC of 0.84. Silveira et al.[50] devised a solution based on gradient boosting machines that focuses primarily on white series analytes. They achieved an AUROC of 0.81 in a dataset composed of anonymous data from 1157 individuals. Banerjee[51] trained both a shallow neural network as well as a random forest model to distinguish COVID-19 cases on data from 954 individuals, reaching an AUROC of 0.94 for those who were admitted to the hospital with severe symptoms, and an AUROC of 0.80 for individuals with mild symptoms. Cabitza et al.[52] evaluate different machine learning algorithms on both a COVID-19 specific dataset as well as another dataset including individuals who exhibited pneumonia symptoms in 2018, consisting of data from 1624 cases. By exploring a variety of biomarkers, including the analytes from CBCs, they were able to achieve an AUROC of 0.90. However, a point of concern for such studies is data scale. We know from the literature that complex machine learning models are prone to overfitting and, with small sample sizes containing only a few hundred individuals, all these works are at risk of presenting unreliable results and overestimated performance.

Wynants et al.[18] provided a study of 37,421 research titles, with 169 studies describing 232 prediction models, of which 208 contained unique, newly developed models. These models contained both a diagnostic solution to identify suspected infection cases as well as prognostic evaluation. One of the key findings was that all models were at high (97%, *n* = 226) or unclear (3%, *n* = 6) risk of bias according to an assessment with PROBAST, suggesting a risk for unreliable predictions when employed in the real world. A similar finding was also reported by Bastos et al.[19], which verified that, out of the 49 risk assessments performed over 5016 references and 40 studies, 98% reported a high risk of individual selection bias. Only 4 studies included outpatients and only two performed some sort of validation at the point of care. This kind of problem is not specific to COVID-19 related research and has been present in many previous medical studies. As mentioned by[53]

*"... failure to proactively and comprehensively mitigate all biases —including latent ones that only emerge over time—risks exacerbating health disparities, eroding public trust in healthcare and health systems, and somewhat ironically, hindering the adoption of AI-based systems that could otherwise help individuals live better lives."*

With that in mind, it is important to highlight the work of Soltan et al.[54] which, with the help of the Oxford University Hospital, included 114,957 individuals in a COVID-negative cohort and 437 in a COVID-positive cohort, thus establishing a dataset of 115,394 individuals for a full study. Before our work, this was the most extensive COVID-19 study to date. While exploring a variety of scenarios regarding COVID-19 prevalence, they reported AUROC values ranging from 0.88 up to 0.94 if their model employs additional data from CBCs, blood gas, and other vital signs collected in routine clinical exams. However, one key concern in this study is the low prevalence of Influenza-like infections (<0.1%), which drew our attention to a different kind of selection bias in COVID-19 research. Due to the hygiene habits acquired by the population worldwide after the pandemic outbreak, we believe that many other confounding diseases might be underrepresented in most performed datasets. As such, models might be learning patterns that are associated with a general infectious condition rather than specifically with COVID-19.

Our concern regarding data bias in the latest COVID-19 research appears to be valid, as was verified during our experiments assessing performance on data before 2020. Several instances of individuals with different variants of the Influenza virus were initially labeled as potential COVID-19 infected, which we knew not to be true. As such, we devised an approach to insert information regarding other diseases into our model without harming accuracy. In particular, we explored two approaches: the first one being simply retraining our specialized model with the added data of negative COVID, whilst keeping positive results for other diseases. The second approach had the objective of creating an ensemble of models with constituents specialized in other virus infections. We observed similar AUROC results between both, with the first one having a slightly higher AUROC result at the cost of lower differentiation capabilities.

We plot the importance of each feature for every individual, and these results are shown in Fig. 7. Yellow points are associated with individuals for whom the corresponding feature shows a relatively high value. Blue points, on the other hand, are associated with individuals for whom the corresponding feature shows a relatively low value. Furthermore, there is a vertical line separating individuals for whom the feature is contributing either to decrease (left side) or increase (right side) the probability of active SARS-Cov-2 infection. Figure 7a shows the COVID-19 specialized model, and the CBC patterns shown in the figure are not specific to COVID-19, as discussed in previous experiments. Figure 7b shows the stacking model, where the COVID-19 specialized model is included as one of the features (i.e., COVID-19 probability). As the stacking model takes into consideration the probability of diverse infections, COVID-19 specific CBC patterns are found.

The stacking approach allows us to study how the physiological patterns found in CBCs of different diseases co-relate. Figure 7c–e illustrates dependence plots of our COVID-19 specialized model prediction concerning remaining diseases, which present relevant patterns that enhance the credibility of our approach. For instance, looking at the right portion of Fig. 7c, we observe a concentration of high Influenza-H1N1 predictions (yellow points) on the upper side of the plot, with a similar pattern on the left side of the plot and a concentration on the lower portion. This behavior shows us that in cases of suspicion of H1N1, the overall

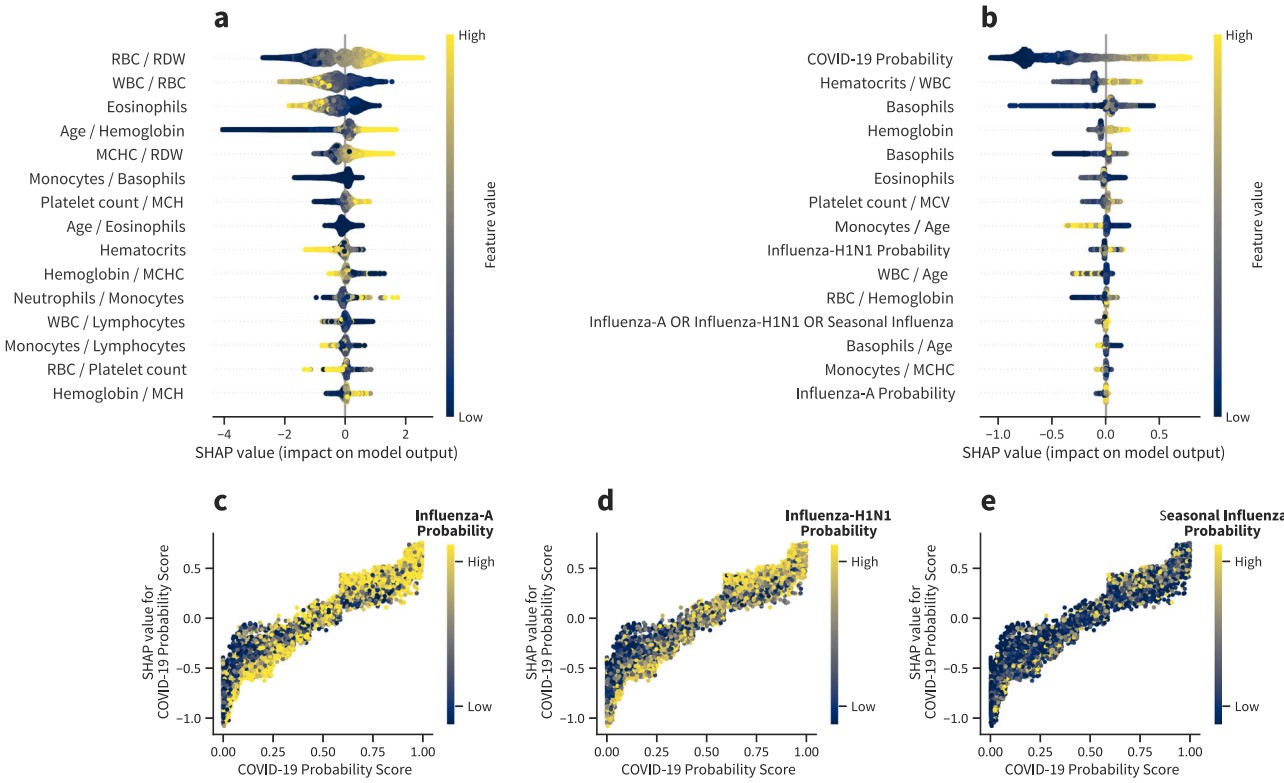

**Fig. 7 Learned analyte patterns and disease prediction relationships. a** SHAP Summary plot for the COVID-19 specialized model. **b** SHAP Summary plot for the stacking model, which combines different specialized models and CBC patterns. **c** Partial dependence plots with the relationship between COVID-19 specialized model's predictions and Influenza-H1N1. **d** Partial dependence plots with the relationship between COVID-19 specialized model's predictions and Influenza-A. **e** Partial dependence plots with the relationship between COVID-19 specialized model's predictions and Seasonal Influenza. $N = 75\,923$ CBCs.

prediction of COVID is significant, be it to confirm an H1N1 hypothesis (left side) or rule it out (right side). However, when there is a lower probability of H1N1, we likewise see a lower scoring attributed to the COVID-19 model. The ensemble learns to use the information regarding all diseases for these hard-to-predict individuals. We observe similar patterns in Influenza-A (Fig. 7d), and Seasonal Influenza (Fig. 7e).

Employing Shapley values as an explanation technique not only allows us to understand the model's final prediction but also to understand the testing time frame. Supplementary Fig. 4 shows a 2D representation of the tests of several individuals contained in the dataset and their respective RT-PCR results for COVID-19. In Supplementary Fig. 4a we observe no clear distinction between exams of infected or healthy individuals and represent what might be observed in an attempt to draw linear correlations between analytes. Supplementary Fig. 4b shows a visualization of the decision process of the model in the shape of a 2D representation of the returned Shapley values. This scenario reflects all the non-linear relationships present in a CBC that might be challenging for humans to extrapolate on their own. Not only can we draw clear divisions between both individual populations but we are also able to infer a measure of confidence. The closer to the decision boundary, the higher the uncertainty of the prediction and, thus, more important the discerning capabilities when combining these results with other relevant factors for diagnosis, such as reported symptoms and possible disease onset period.

Predicting data from the second wave proved to be particularly hard, as we observed a deterioration in the performance of our first wave model as time went on, which might be associated with concept drift. In particular, we observed that the peak in performance on the second half of the chart is associated with a lower COVID-19 prevalence, which implied that the model was

losing its ability to predict COVID-19 infections. We hypothesize some explanations for this behavior, including the effect of distribution of COVID-19 prevalence in 2020 and across 2021, as well as the prevalence of other possible confounding diseases, which changed as restriction measures were lifted. Likewise, one of the main characteristics of the second wave is the emergence of a new COVID-19 strain, namely the P.1 variant that ran rampant in Brazil during the analyzed period. It might be the case that the physiological reaction of the body to the new strain was distinct from the earlier variants, resulting in degradation in performance. Finally, another possibility is that RT-PCR tests at the time of evaluation might not have been tuned to properly identify the new strain, thus inducing a divergence between model output and ground-truth data due to possible false negatives.

The proposed solution consisted of employing data closer to the start of the second wave, simulating a scenario where we keep the model as up-to-date as possible before the start of a new pandemic stage. Although we could not test for each of these hypotheses, the proposed approach should solve for all of the three possible explanations described. With this approach, not only did we verify a higher performance from the start, but the model was able to largely mitigate the concept drift phenomena, retaining an AUROC above the 0.90 threshold throughout most of the evaluated period.

A point of attention that should be addressed by any health professional when employing our approach is the presence of co-infections. For instance, multiple cases of COVID-19 hospital cross-infections have been identified[55]. As we do not have data explicitly concerning co-infections, we cannot provide insights regarding the blood profiles that emerge in such situations which might confuse the model. It is also important to highlight the impact of ethnicity on CBC results[56]. Although

the large data sample and the demographic plurality of Brazil serve as indicators of robustness, further testing is needed to understand if the Brazilian model can be directly applied to other contexts. Nevertheless, our method is generalist to an extent that the achieved results could be potentially replicated anywhere on Earth if data concerning a specific region/scenario is collected.

In this work, we proposed a novel machine intelligence approach to automated COVID-19 diagnosis through complete blood counts, a repurposing of an accessible and low-cost exam. The task was formulated as a binary classification problem to predict which analyte combinations are likely to be associated with SARS-Cov-2 infection. We evaluated our approach on a dataset containing over a million exams which, to the author's knowledge, is the largest COVID-19 dataset to date. One of our key results pointed out that training machine learning models solely on 2020 data are not enough to guarantee robustness in real-world applications even with high reported performance estimates. This raises several concerns regarding the latest COVID-19 machine learning literature and confirms issues that were already brought to attention but not properly addressed. Providing information regarding other diseases was essential to guarantee robustness and our stacking approach, which presented a high performance in the wake of scenarios with both prevalence and absence of COVID-19 infections, with a reported AUROC of 0.90+.

For future work, it is imperative to assess the impact of our approach on a hospital's daily flow, as the adoption of new technology can potentially disrupt existing processes. This should also enable us to collect data concerning other relevant analysis, such as co-infections and studying the impact of different demographic profiles. We are currently implementing the developed algorithm in different Brazilian hospitals using an API framework connected directly to their databases. In these scenarios, we aim to understand how the proposed tool can be introduced into a hospital's existing workflow in the least disruptive way, as well as find out how comfortable health professionals feel when using it. Further validation thesis include observing health professionals' interactions with the tool and possible changes in procedures, protocols, and decision-making processes, as well as the benefits of the solution if applied in fast-paced and high-volume contexts.

Since CBCs are widely available and provide results at a fast pace, different use cases have been mapped: to potentially speed up triaging processes in hospitals where other forms of diagnosis are tardier; to support clinical diagnosis and triaging in hospitals where other forms of diagnosis are scarce or unavailable; and to reduce overall system burdening of traditional diagnostic methods when applied as pre-tests to non-emergency situations (such as elective surgeries), to name a few examples. Given its versatile nature, low cost, and speed, we believe our tool to be particularly useful in a variety of scenarios—both pandemic and post-pandemic.

## Data availability

All source data for the figures in the main manuscript and the Supplementary Information available for non-commercial use have been deposited at https://doi.org/10.6084/m9.figshare.1504679[57]. Additional datasets were used in this study under specific conditions and are restricted due to confidentiality limitations. Requests to access additional datasets should be directed to wesley.prieto@grupofleury.com and will undergo internal approval from Grupo Fleury.

## Code availability

The code used for the machine-learning analyses available for non-commercial use has been deposited at https://doi.org/10.6084/m9.figshare.1504679[57].

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

## Author contributions

G.Z., A.V., W.H.P., and C.F.H.G. conceptualized this study and conceived the experiments, G.Z., A.V., and D.A. conducted the experiments, data curation and software development, G.Z., A.V., V.R., M.G.S., W.H.P., M.C.P., C.S.L., and C.F.H.G. worked on the writing and original draft preparation. All authors reviewed the manuscript.

## Competing interests

The authors declare no competing interests.
