## [Peer Review File · Communications Medicine]

Reviewers' comments:

Reviewer #1 (Remarks to the Author):

The paper is interesting and presents a work that has a clear potential to interest and inform a wide readership. However, the quality is not homogeneous through the sections and I believe that there are some parts that can be improved to get an excellent work.

In what follows, I followed the IJMEDI checklist, which was suggested to me by a knowledgeable colleague in ML (<https://zenodo.org/record/4835800#.YS-m8t-xVhE>) and that I find appropriate for medical informatics applications [6].

Thus, by roughly following the above resource, I share the following comments with the authors.

1) The pre-processing part of the study seems to be very well done. I appreciated that the authors took great care in avoiding possible sources of bias, e.g. due to age, gender or the presence of false-negatives in RT-PCR. However, the authors should report about the distribution of the main predictive features in the used dataset (e.g., mean and standard deviation for continuous variables) and the demographics distribution of the study population (e.g. ethnic groups, age distribution). Moreover, other data about the patient population would be valuable: for instance, were the patients admitted in emergency room? hospitalized? Were they symptomatic or also not symptomatic? Moreover, I wonder if this model is generalizable also on children (the infection disease have a different prevalence according to the age): in any case, the authors should be more clear on the scope and applicability of their model.

2) a related but more technical point: more information is definitely needed about the hematological features! Laboratory data results depend on the instrument used / analytical method /reagent and they can be expressed in different units of measurements. This information cannot be missing in a study using laboratory data.

Over the past decade, laboratory medicine official bodies and scientific societies have highlighted the importance of standardization and harmonization, and the International Federation of Clinical Chemistry and Laboratory Medicine has recommended: "To ensure unambiguity in reporting values, only one expression for a unit of a given magnitude should be used" [1].

The harmonization concept deserves to be clarified. For optimal laboratory service, results from different measurement procedures (MPs) for the same measurand should be equivalent (harmonized) within stated specifications, enabling the results to be used reliably for medical decisions and to reduce the risk for erroneous interpretations based on test results. The term "harmonization" refers to any process that enables establishing equivalence of reported values among different end-user MPs. Harmonization of results among different MPs is an essential characteristic when using medical decision values from clinical practice guidelines. Harmonized results are also required to enable use of common reference intervals for different MPs when decision values based on clinical [2].

For the CBC measurements, it is to be noted that it is a multiparameter analysis, where only a minority of parameters (hemoglobin (Hb), white blood cell (WBC), red blood cells (RBC), mean corpuscular volume (MCV), platelets (PLT), and reticulocytes (RET)) are simultaneously and directly measured by different instrumental channels. Moreover, hematological analyzers may use different measurement principles (optical, impedance, fluorescence) to detect cellular characteristics [3-5]. The principle of the instrument is a requirement to publish CBC results especially for some measurands (platelet can significantly change if obtained from a method base on impedance or fluorescence)

That said, the following pieces of information are missing and should be added (in a table)

2.1) Are this data from the same instrument (manufacturer and model of the instrument)?

2.2) Are this data obtained from the same analytical principle (optical, impedance, fluorescence)?

2.3) What are the unit of measurements considered? The subpopulation can be expressed as % values of the total WBC, or absolute values

2.4) Particularly the red cells related parameters depend on the sex (they have also different normal values in male and female population) and they should be considered differently

2.5) What is the clinical rationale to combine features like age and eosinophils? PLT e MCH? RBC e PLT? It is evident the rationale used for RBC and RDW, however other unions seem casual. Authors should explain this rationale.

3) Was the data impacted by missing data? If this was the case, did the authors apply any imputation technique or other management strategies?

4) Did the authors perform any feature preprocessing (e.g. standardization, normalization)? By the way, the approach to feature selection that the authors adopted, based on graph search using the A* algorithm, seems an interesting approach that is not frequently seen in the literature: well done!

5) The authors report to have used lightGBM as the chosen ML model. Which hyper-parameter settings did they employ (e.g. number of estimators)? Apart from feature selection, did the authors apply any hyper-parameter optimization?

6) The proposed stacking approach seems quite effective, since as seen in Fig. 7 the number of truly negative patients predicted to be negative increases w.r.t. the specialized COVID-19 model, and, more in general, the trained model shows a good AUC both in the cross-validation and the validation sets.

Nonetheless, since the authors compare the results of different models, they should performed statistical hypothesis testing to assess the presence of statistically significant differences (if any), with appropriate corrections for multiple testing.

7) The author provide a basic report about model calibration in Figure 8, albeit it may be interesting if the authors reported the full calibration/reliability curve for the developed model.

8) Also and related to the above point, it would be interesting to see the main performance metrics (i.e. specificity, sensitivity, accuracy, AUROC) for the developed models, and not only for the prevalence simulation (see Table 2).

If the authors address all the above 8 points (and others that they could consider by reviewing the IJMEDI checklist) I will have no problem recommending this work for publication (please in the response letter refer to each point explicitly). Thank you.

References

1) Mussap M. Clinical Laboratory Test Unit Homogeneity—an Urgent Need. *JAMA Intern Med.* 2020;180(12):1715–1716. doi:10.1001/jamainternmed.2020.3532

2) Miller WG, Greenberg N. Harmonization and Standardization: Where Are We Now? *J Appl Lab*

Med. 2021 Mar 1;6(2):510-521. doi: 10.1093/jalm/jfaa189. PMID: 33241270.

3) Vidali M, Carobene A, Apassiti Esposito S, Napolitano G, Caracciolo A, Seghezzi M, Previtali G, Lippi G, Buoro S. Standardization and harmonization in hematology: Instrument alignment, quality control materials, and commutability issue. *Int J Lab Hematol*. 2021 Jun;43(3):364-371

4) Buttarello M, Plebani M. Automated blood cell counts: state of the art. *Am J Clin Pathol*. 2008 Jul;130(1):104-16. doi: 10.1309/EK3C7CTDKNVPXVTN. PMID: 18550479.

5) Verbrugge SE, Huisman A. Verification and standardization of blood cell counters for routine clinical laboratory tests. *Clin Lab Med*. 2015 Mar;35(1):183-96. doi: 10.1016/j.cll.2014.10.008

6) Cabitza, F., & Campagner, A. (2021). The need to separate the wheat from the chaff in medical informatics. *Int J Med Inform*. 2021 Sep;153:104510. doi: 10.1016/j.ijmedinf.2021.104510.

Reviewer #2 (Remarks to the Author):

The article outlines the use of CBC in determining SARS cov2 infection and its limitations compared to rtPCR. A strength of this paper is its investigation using data from different patient infections and its conclusions are sound and novel.

Training and validation sets are of large size, and the first and second wave data are compared.

A definition of what constitutes the first and second wave is needed in the table or in the legend (its outlined on p6).

The manner by which sets of data are excluded in order to remove false positives benefits the paper – is this the first example of this or should this be referenced?

I am unable to make comment on the maths involved since this is not my profession.

Reviewer #3 (Remarks to the Author):

The authors applied a machine intelligence approach (i.e. a stacking procedure which optimally combines models for different respiratory infections) to develop a model which automates COVID-19 diagnosis through CBCs. This work was done on a big COVID-19 related dataset containing over a million exams. The final model reached .914 AUROC in in-sample cross-validation and .917 AUROC in out-of-sample validation. Thus, this CBC-based model is reliable to diagnose COVID-19 in the presence of other respiratory diseases in real-world conditions. This is a nicely preformed work with an exhaustive methodology which addresses an important question. However, I have a few comments and concerns:

1.As CBC is an accessible and low-cost exam, it would be better for the authors to discuss how their model could be easily implemented in daily clinical settings to help COVID-19 diagnosis and management.

2.Marked changes in CBC parameters were observed near the day when the first positive RT-PCR was shown. Could the authors elaborate the possible reasons for it since this would apparently affect the results. Also, the performance of the First-wave model deteriorates with time, could the authors give more detailed explanation on that?

3.Flowcharts may help to depict the processes of sample selection, variable selection, and model development and validation more clearly and easier to understand. Also, whether there are missing

data, and if yes, how they were treated?

4. Generalisability and applicability of a model to different cohorts is a key metric in deciding its clinical utility. Given the differences of health care systems, characteristics of the affected populations (e.g., races), and ways of COVID-19 affected different populations (e.g., variants of the virus) among countries, it would be interesting if the authors validate their model in other datasets outside Brazil to generalize the use of the model.

5. Several factors, such as age, gender, and pre-existing co-morbidities may influence the capacity of the model to diagnose COVID-19. Could the authors comment on whether the model should be interpreted differently in patients with these different levels of confounders?

Belo Horizonte, Brazil, October 8, 2021

We would like to thank once again the editor and reviewers for their interest in our work as well as their comments and suggestions, which we considered to improve our manuscript. All the modified passages are marked in red in the revised manuscript. In this document, we answer and discuss all the points raised by the reviewers.

Reviewer 1

The paper is interesting and presents a work that has a clear potential to interest and inform a wide readership. However, the quality is not homogeneous through the sections and I believe that there are some parts that can be improved to get an excellent work.

In what follows, I followed the IJMEDI checklist, which was suggested to me by a knowledgeable colleague in ML (<https://zenodo.org/record/4835800>) and that I find appropriate for medical informatics applications.

Thus, by roughly following the above resource, I share the following comments with the authors.

Reviewer point P.1.1: The pre-processing part of the study seems to be very well done. I appreciated that the authors took great care in avoiding possible sources of bias, e.g. due to age, gender or the presence of false-negatives in RT-PCR. However, the authors should report about the distribution of the main predictive features in the used dataset (e.g., mean and standard deviation for continuous variables) and the demographics distribution of the study population (e.g. ethnic groups, age distribution). Moreover, other data about the patient population would be valuable: for instance, were the patients admitted in emergency room? hospitalized? Were they symptomatic or also not symptomatic? Moreover, I wonder if this model is generalizable also on children (the infection disease have a different prevalence according to the age): in any case, the authors should be more clear on the scope and applicability of their model.

Reply: We thank the reviewer for the pertinent remarks. Regarding the distribution of the main predictive features, we included Table 2 on page 6 containing the means and standard deviation of each feature. Figure 2 should complement this Table by presenting the age distributions in each of these cohorts as histograms. Concerning the additional data about the patient population, unfortunately, our dataset only contains laboratory data as that is one of Fleury Group's main activities. Thus, we were not able to investigate demographics strata aside from sex and age or clinical and symptomatic strata. Our intention was to discover a robust solution to support medical decision in a hospital setting for patient screening using existing/readily available data to ensure a quick implementation and increase the impact of our findings. We plan to study these demographic aspects in-depth in future work once we implement our approach in a hospital, and we added a passage briefly describing these goals in the Conclusion, page 15 lines 374-387. We also delineate the exact scope of our work given the characteristics of our data in the Introduction, page 2 lines 41-42, and in the footnote of the same page, hoping that it is clearer now.

Reviewer point P.1.2: A related but more technical point: more information is definitely needed about the hematological features! Laboratory data results depend on the instrument used / analytical method /reagent and they can be expressed in different units of measurements. This information cannot be missing in a study using laboratory data.

Over the past decade, laboratory medicine official bodies and scientific societies have highlighted the importance of standardization and harmonization, and the International Federation of Clinical Chemistry and Laboratory Medicine has recommended: “To ensure unambiguity in reporting values, only one expression for a unit of a given magnitude should be used”.

The harmonization concept deserves to be clarified. For optimal laboratory service, results from different measurement procedures (MPs) for the same measurand should be equivalent (harmonized) within stated specifications, enabling the results to be used reliably for medical decisions and to reduce the risk for erroneous interpretations based on test results. The term “harmonization” refers to any process that enables establishing equivalence of reported values among different end-user MPs. Harmonization of results among different MPs is an essential characteristic when using medical decision values from clinical practice guidelines. Harmonized results are also required to enable use of common reference intervals for different MPs when decision values based on clinical.

For the CBC measurements, it is to be noted that it is a multiparameter analysis, where only a minority of parameters (hemoglobin (Hb), white blood cell (WBC), red blood cells (RBC), mean corpuscular volume (MCV), platelets (PLT), and reticulocytes (RET)) are simultaneously and directly measured by different instrumental channels. Moreover, hematological analyzers may use different measurement principles (optical, impedance, fluorescence) to detect cellular characteristics.

The principle of the instrument is a requirement to publish CBC results especially for some measurands (platelet can significantly change if obtained from a method base on impedance or fluorescence)

That said, the following pieces of information are missing and should be added (in a table)

Reviewer point P.1.2.1: Are this data from the same instrument (manufacturer and model of the instrument)?

Reply: The reviewer is correct in that it is imperative to follow harmonization and standardization procedures to ensure unambiguity. We do follow strict quality control and perform harmonization of equipment according to the Clinical and Laboratory Standards Institute (CLSI) guideline. All CBC measurements were obtained from EDTA-K3 collected peripheral blood samples analyzed by the Automated Hematology Analyzer XT or XN series from Sysmex (Sysmex Corporation, Kobe, Japan). We added a passage on Pages 3 and 4, lines 117-130 describing harmonization, the measurement procedures, and equipment. We also took the liberty of adding one of the referenced works into our bibliography. We thank the reviewer for raising awareness of this issue.

Reviewer point P.1.2.2: Are this data obtained from the same analytical principle (optical, impedance, fluorescence)?

Reply: The same analytical principle was used to obtain the data. This is now described on page 5 lines 139-141 of the revised manuscript.

Reviewer point P.1.2.3: What are the unit of measurements considered? The subpopulation can be expressed as % values of the total WBC, or absolute values

Reply: We apologize for this, Table 2 was added on page 6 of the revised manuscript addressing the units of measure of each evaluated analyte, as well as the observed means and standard deviations in each cohort.

Reviewer point P.1.2.4: Particularly the red cells related parameters depend on the sex (they have also different normal values in male and female population) and they should be considered differently

Reply: You are completely correct. To take into account the different analyte levels that are appropriate to males and females, we normalize all analytes by their respective reference values given sex and age. This enables us to compare CBCs of patients of different sex and ages in the same model. A passage describing this in more detail was added to **Data and Models** section, page 5, lines 150-165.

Reviewer point P.1.2.5: What is the clinical rationale to combine features like age and eosinophils? PLT e MCH? RBC e PLT? It is evident the rationale used for RBC and RDW, however other unions seem casual. Authors should explain this rationale.

Reply: We appended a passage discussing the rationale regarding this sort of feature expansion on page 5, lines 166-176. Briefly, we assume that there exist unknown ratios that are yet to be studied in-depth. We allow the algorithm to search all possible ratio combinations given this possible gap in the literature and avoid spurious correlations by extensive cross-validations. If a given feature consistently presents itself as useful after evaluation on the many proposed scenarios with 5-fold cross-validation, we consider this as a strong indicator of its predictive power. Feature importance is asserted through shap values, and we enforce a strictly non-zero impact on model output assuming a 95% confidence interval.

Reviewer point P.1.3: Was the data impacted by missing data? If this was the case, did the authors apply any imputation technique or other management strategies?

Reply: Given the size of our dataset and that we employed only analytes associated with routine exams encompassed by the CBC, we did not work with missing data regarding the input features themselves. The only instances of missing data are associated with labeling exam pairs that were taken at different time intervals, of which we address in section **Safe Labeling**.

Reviewer point P.1.4: Did the authors perform any feature preprocessing (e.g. standardization, normalization)? By the way, the approach to feature selection that the authors adopted, based on graph search using the A* algorithm, seems an interesting approach that

is not frequently seen in the literature: well done!

Reply: We thank the reviewer for the compliments and are glad that you approved of our approach. Regarding feature preprocessing, we perform normalization using the reference values as a pivot. As mentioned in P.1.2.4, we discuss this in detail on **Data and Models** section of the revised manuscript, page 5, lines 150-158.

Reviewer point P.1.5: The authors report to have used lightGBM as the chosen ML model. Which hyper-parameter settings did they employ (e.g. number of estimators)? Apart from feature selection, did the authors apply any hyper-parameter optimization?

Reply: We perform an exhausting grid search to find a suitable set of hyperparameters optimizing the cross-entropy loss function. We added a passage describing the optimal hyperparameters found in section **Model Training**, page 8 lines 229-231.

Reviewer point P.1.6: The proposed stacking approach seems quite effective, since as seen in Fig. 7 the number of truly negative patients predicted to be negative increases w.r.t. the specialized COVID-19 model, and, more in general, the trained model shows a good AUC both in the cross-validation and the validation sets. Nonetheless, since the authors compare the results of different models, they should performed statistical hypothesis testing to assess the presence of statistically significant differences (if any), with appropriate corrections for multiple testing.

Reply: For all the results we assess statistical significance through paired t-test with a two-sided 95% confidence interval. We made this explicit by adding a passage in section **Experimental Design** on page 9, lines 254-255. We would like to thank the reviewer for pointing this out.

Reviewer point P.1.7: The author provide a basic report about model calibration in Figure 8, albeit it may be interesting if the authors reported the full calibration/reliability curve for the developed model.

Reply: We replaced Figure 8 (now Figure 9) on Page 13 from a heatmap to the full curve, providing a more extensive histogram of predictions as well as the underlying PDF and CDF curves. We hope that the result becomes more clear in this manner.

Reviewer point P.1.8: Also and related to the above point, it would be interesting to see the main performance metrics (i.e. specificity, sensitivity, accuracy, AUROC) for the developed models, and not only for the prevalence simulation (see Table 2).

Reply: We added all the relevant metrics to all evaluated scenarios, now present on page 2 line 49, page 9 line 265, page 10 figure 4, and page 11 line 297.

Reviewer 2

The article outlines the use of CBC in determining SARS cov2 infection and its limitations compared to rtPCR. A strength of this paper is its investigation using data from different patient infections and its conclusions are sound and novel.

Reviewer point P.2.1: Training and validation sets are of large size, and the first and second wave data are compared. A definition of what constitutes the first and second wave is needed in the table or in the legend (its outlined on p6).

Reply: We apologize for this issue. The reviewer is correct that this definition is introduced too late into the manuscript. We added a passage to our introduction on Page 2, lines 32-35, stating what we believe to be the beginning of the Second Wave of Covid in Brazil. We also included this in the description of Table 2 with the considered split points between the first and second waves.

Reviewer point P.2.2: The manner by which sets of data are excluded in order to remove false positives benefits the paper – is this the first example of this or should this be referenced?

Reply: We are pleased that the reviewer approved of our safe labeling step. To the authors' knowledge, this is the first example of this sort of approach. We are not aware of any related works that filter possible false positives or negatives like ours.

Reviewer 3

The authors applied a machine intelligence approach (i.e. a stacking procedure which optimally combines models for different respiratory infections) to develop a model which automates COVID-19 diagnosis through CBCs. This work was done on a big COVID-19 related dataset containing over a million exams. The final model reached .914 AUROC in in-sample cross-validation and .917 AUROC in out-of-sample validation. Thus, this CBC-based model is reliable to diagnose COVID-19 in the presence of other respiratory diseases in real-world conditions. This is a nicely preformed work with an exhaustive methodology which addresses an important question. However, I have a few comments and concerns:

Reviewer point P.3.1: As CBC is an accessible and low-cost exam, it would be better for the authors to discuss how their model could be easily implemented in daily clinical settings to help COVID-19 diagnosis and management.

Reply: We are currently at the stage of implementing our approach in hospitals that are partners with the Fleury Group, and one of our main goals in future work is to assess how

can it be integrated into a hospital’s daily routine to be less disruptive as possible. Currently, we are working with an approach devised to connect an API to the hospital’s database and, as soon as the CBC result is ready, the diagnostician could request the AI prediction in a mostly seamless way. We believe that the main benefited sector should be on the triage stage, and we will validate this hypothesis during our roadmap. We included this discussion of our next goals in the Conclusion, page 15 lines 374-387.

Reviewer point P.3.2: Marked changes in CBC parameters were observed near the day when the first positive RT-PCR was shown. Could the authors elaborate the possible reasons for it since this would apparently affect the results. Also, the performance of the First-wave model deteriorates with time, could the authors give more detailed explanation on that?

Reply: Our data covers patients who went to one of the Fleury Group laboratories to undergo an rtPCR exam, as well as all the CBCs from these patients. We hypothesize that the search for an rtPCR exam, in particular for patients who obtained a confirmatory diagnosis of COVID-19, is associated with symptom onset. We believe that a patient’s first positive rtPCR result is also associated with the infectious peak, thus presenting a distinct pattern on an individuals’ physiology. Evidently, this is only an approximation as we have no access to the actual disease progress for each patient.

As seen in Figure 1, we observe that the blood cells values do not remain constant during the whole disease timeframe. For instance, we observed a pattern of a drop in WBC near the first positive rtPCR result, followed by a sharp increase in its count for the next two weeks before eventually returning to reference values, albeit slowly. We believe that these effects are associated not only with the evolution of the inflammatory stage of COVID-19 but possibly with the effect of medications and treatments used to combat the disease. Since most of these are hypotheses, we opted to focus our analysis around the first positive rtPCR result, being as close to the peak of the infectious phase as possible.

Regarding performance deterioration, we believe this to be associated with the emergence of new covid-19 strains, in particular the P.1 variant that ran rampant in Brazil during the evaluated period. It might be the case that the physiological reaction of the body to the new strain is distinct from the earlier variants, thus resulting in degradation in performance. This can be remedied by training with as up-to-date data as possible. It can also be the case that the rtPCR exams were not properly identifying the new strain, thus presenting possible false negatives for it. Once again, retraining the model allows it to properly tune itself to give predictions following whatever the rtPCR specifications are at the time.

Reviewer point P.3.3: Flowcharts may help to depict the processes of sample selection, variable selection, and model development and validation more clearly and easier to understand. Also, whether there are missing data, and if yes, how they were treated?

Reply: We included Figure 3 on Page 9 of the revised manuscript. It depicts the flowchart of our learning pipeline from data pre-processing to final prediction. We hope that with this our approach becomes more clear and easy to understand. Given the size of our dataset, we were able to avoid missing data. The only uncertainty lies with the labeling of exam pairs that were taken at different time intervals from one another, of which we address in section

Safe Labeling.

Reviewer point P.3.4: Generalisability and applicability of a model to different cohorts is a key metric in deciding its clinical utility. Given the differences of health care systems, characteristics of the affected populations (e.g., races), and ways of COVID-19 affected different populations (e.g., variants of the virus) among countries, it would be interesting if the authors validate their model in other datasets outside Brazil to generalize the use of the model.

Reply: You are correct in these assertions and indeed these analyses would prove to be of utmost interest. In this work, we focused on understanding the Brazilian scenario, but we plan to extend our research to other countries and other subpopulations in future work. We thank the reviewer for the suggestion.

Reviewer point P.3.5: Several factors, such as age, gender, and pre-existing co-morbidities may influence the capacity of the model to diagnose COVID-19. Could the authors comment on whether the model should be interpreted differently in patients with these different levels of confounders?

Reply: As the reviewer properly mentioned, the model does not have access to full patient data such as co-morbidities or exam history. This is mostly because this information is not present in our working dataset, although an assortment of patient profiles should be in place. However, we do not wish for our model to be a replacement of any kind to doctors and physicians. We hoped to devise a tool that could aid health professional's decision-making. As such, we propose that our model should be employed in conjunction with whatever information the health care provider has access to at the moment. Just as a physician might request an additional chest x-ray even after a negative rtPCR result, we believe that our model could be an additional tool to aid a diagnostician.

Reviewers' comments:

Reviewer #1 (Remarks to the Author):

I thank the authors because they have addressed all of the points I had raised with adequate (the statistical concerns) and improvable effectiveness (the laboratory medicine points). Since I believe that the manuscript will be a much better contribution to the literature thanks to these comments (as well as the comments of other reviewers), I urge authors to clearly cite the guideline I have suggested in their work, saying that they followed the recommendations of the IJMEDI checklist. This would make more visible an open-source contribution that also in their case significantly improved their work and thus it would spread more appropriate and common reporting practices in the community to improve the interpretation and reproducibility of the results obtained.

The correct reference is: Cabitza F, Campagner A. The need to separate the wheat from the chaff in medical informatics: Introducing a comprehensive checklist for the (self)-assessment of medical AI studies. *Int J Med Inform.* 2021 Sep;153:104510. doi: 10.1016/j.ijmedinf.2021.104510. Epub 2021 Jun 2. PMID: 34108105.

In addition, what the authors definitely need to fix is Table 2. Almost every entry there needs to be amended to make it conform to international guidelines or the "new" standards of reporting in laboratory medicine. Below, I will give a few examples, indicating, first, the incorrect entry and, then, its rewriting according to good reporting practices. Although the authors could find converting values into other units of measure overly pedantic, I think these elements are not secondary: in fact, using correct units of measurement facilitates reproducibility of results and evaluation of results. Once the above guidelines were referenced correctly and Table 2 was reviewed systematically, the article could be published and be a good contribution to the specialist literature. Thank you.

ERRATUM

RBC (millions/mm³) 5.06 ± 0.52

CORRIGENDUM

RBC (10¹²/L) 5.06 ± 0.52

ERRATUM

Hemoglobin (g/dl) 14.89 ± 1.40

CORRIGENDUM

Hemoglobin (g/dl) 14.9 ± 1.4

ERRATUM

Hematocrit (%) 43.75 ± 3.96

CORRIGENDUM

Hematocrit (%) 43.8 ± 4.0

ERRATUM

MCV (fL) 86.75 ± 4.75

CORRIGENDUM

MCV (fL) 86.8 ± 4.8

ERRATUM

MCH (pg) 29.52 ± 1.88

CORRIGENDUM

MCH (pg/cell) 29.5 ± 1.9

ERRATUM

MCHC (g/dL) 34.07 ± 1.07

CORRIGENDUM

MCHC (g/dL) 34.1 ± 1.1

ERRATUM

RDW (%) 12.99 ± 1.01

CORRIGENDUM

RDW (%) 13.0 ± 1.0

ERRATUM

WBC (/mm³) 6069.27 ± 2368.75

CORRIGENDUM

WBC (10⁹/L) 6.07 ± 2.37

ERRATUM

Monocytes (/mm³) 662.69 ± 292.48

CORRIGENDUM

Monocytes (10⁹/L) 0.66 ± 0.29

ERRATUM

Lymphocytes (/mm³) 1399.01 ± 721.64

CORRIGENDUM

Lymphocytes (10⁹/L) 1.40 ± 0.72

ERRATUM

Eosinophils (/mm³) 70.53 ± 95.87

CORRIGENDUM

Eosinophils (10⁹/L) 0.71 ± 0.10

ERRATUM

Basophils (/mm³) 21.85 ± 16.44

CORRIGENDUM

Basophils (10⁹/L) 0.22 ± 0.16

ERRATUM

Neutrophils (/mm³) 3911.75 ± 2222.41

CORRIGENDUM

Neutrophils (10⁹/L) 3.91 ± 2.22

ERRATUM

Platelets (1,000/mm³) 195.66 ± 56.72

CORRIGENDUM

Platelets (10⁹/L) 195.7 ± 56.7

Reviewer #2 (Remarks to the Author):

The questions raised have been addressed sufficiently for publication

Reviewer #3 (Remarks to the Author):

Comments for COMMSMED-12-0341A: The article applied a machine learning models (ie, a stacking program that optimally combines different respiratory infection models) to develop a model that can automatically diagnose COVID-19 through CBC. The final model achieved 0.914 AUROC cross validation in the sample and 0.917 AUROC for out-of-sample validation. Therefore, this CBC-based model is a reliable condition for diagnosing COVID-19 when there are other respiratory diseases in the real world. This is a very interesting work. I still have some comments and concerns.

1.Generalisability and applicability of a model to different cohorts is a key metric in deciding its clinical utility.the should report about the demographics distribution of the study population (e.g. ethnic groups, age distribution). Moreover, other data about the patient population would be valuable: for instance, were the patients admitted in emergency room? hospitalized?

2.Should be more clear on the scope and applicability of their model. Can it be applied to countries other than Brazil

3.Interestingly, significant changes in CBC parameters were observed on the day of the first RT-PCR positive. Can this phenomenon be observed in all data sets? Author should discuss the possible reasons.

4.Abnormal white blood cell values may be a sign of various infections, and it is not uncommon for patients with new coronavirus to have other respiratory virus, bacterial, and fungal infections. You should consider the possible impact of co-infection on the applicability of the model.

5.The performance of the First-wave model deteriorates over time. Can the author give a more detailed explanation? Please discuss.The experiment emphasized the importance of retraining the models so that they can explain the virus mutation. Affected the applicable value of the model ?

6.No ethical information being given.

7. According to the IJMEDI checklist for assessment of medical AI, there is a lot of unreachable information in your data set, you should discuss the limitations

Comments for COMMSMED-12-0341A: The article applied a machine learning models (ie, a stacking program that optimally combines different respiratory infection models) to develop a model that can automatically diagnose COVID-19 through CBC. The final model achieved 0.914 AUROC cross validation in the sample and 0.917 AUROC for out-of-sample validation. Therefore, this CBC-based model is a reliable condition for diagnosing COVID-19 when there are other respiratory diseases in the real world. This is a very interesting work. I still have some comments and concerns.

1. Generalisability and applicability of a model to different cohorts is a key metric in deciding its clinical utility. The should report about the demographics distribution of the study population (e.g. ethnic groups, age distribution). Moreover, other data about the patient population would be valuable: for instance, were the patients admitted in emergency room? hospitalized?

2. Should be more clear on the scope and applicability of their model. Can it be applied to countries other than Brazil

3. Interestingly, significant changes in CBC parameters were observed on the day of the first RT-PCR positive. Can this phenomenon be observed in all data sets? Author should discuss the possible reasons.

4. Abnormal white blood cell values may be a sign of various infections, and it is not uncommon for patients with new coronavirus to have other respiratory virus, bacterial, and fungal infections. You should consider the possible impact of co-infection on the applicability of the model.

5. The performance of the First-wave model deteriorates over time. Can the author give a more detailed explanation? Please discuss. The experiment emphasized the importance of retraining the models so that they can explain the virus mutation. Affected the applicable value of the model?

6. No ethical information being given.

7. According to the IJMEDI checklist for assessment of medical AI, there is a lot of unreachable information in your data set, you should discuss the limitations

Belo Horizonte, Brazil, December 21, 2021

We would like to thank once again the editor and reviewers for their interest in our work as well as their comments and suggestions, which we considered to improve our manuscript. All the modified passages are marked in red in the revised manuscript. In this document, we answer and discuss all the points raised by the reviewers.

Reviewer 1

I thank the authors because they have addressed all of the points I had raised with adequate (the statistical concerns) and improvable effectiveness (the laboratory medicine points).

Reviewer point P.1.1: Since I believe that the manuscript will be a much better contribution to the literature thanks to these comments (as well as the comments of other reviewers), I urge authors to clearly cite the guideline I have suggested in their work, saying that they followed the recommendations of the IJMEDI checklist. This would make more visible an open-source contribution that also in their case significantly improved their work and thus it would spread more appropriate and common reporting practices in the community to improve the interpretation and reproducibility of the results obtained.

The correct reference is: Cabitza F, Campagner A. *The need to separate the wheat from the chaff in medical informatics: Introducing a comprehensive checklist for the (self)-assessment of medical AI studies*. Int J Med Inform. 2021 Sep;153:104510. doi: 10.1016/j.ijmedinf.2021.104510. Epub 2021 Jun 2. PMID: 34108105.

Reply: The Reviewer is completely correct and we wish to both apologize for this hindsight and thank you once again for directing us to the IJMEDI checklist. The aforementioned reference was added to the manuscript on page 2, lines 42-43.

Reviewer point P.1.2: In addition, what the authors definitely need to fix is Table 2. Almost every entry there needs to be amended to make it conform to international guidelines or the "new" standards of reporting in laboratory medicine. Below, I will give a few examples, indicating, first, the incorrect entry and, then, its rewriting according to good reporting practices. Although the authors could find converting values into other units of measure overly pedantic, I think these elements are not secondary: in fact, using correct units of measurement facilitates reproducibility of results and evaluation of results. Once the above guidelines were referenced correctly and Table 2 was reviewed systematically, the article could be published and be a good contribution to the specialist literature. Thank you.

Reply: We thank the reviewer for the zeal and for providing examples for the correct standards for each input variable. It is indeed imperative to provide sensible units of measure to make the work both more reproducible and facilitate its understanding. We properly fixed Table 2 with the correct units of measure and the number of precision digits.

Reviewer 2

The questions raised have been addressed sufficiently for publication

Reply: We are pleased that our work met the Reviewer's standards and would like to thank you once again for the helpful insights provided so far.

Reviewer 3

The article applied a machine learning models (ie, a stacking program that optimally combines different respiratory infection models) to develop a model that can automatically diagnose COVID-19 through CBC. The final model achieved 0.914 AUROC cross validation in the sample and 0.917 AUROC for out-of-sample validation. Therefore, this CBC-based model is a reliable condition for diagnosing COVID-19 when there are other respiratory diseases in the real world. This is a very interesting work. I still have some comments and concerns.

Reviewer point P.3.1: Generalisability and applicability of a model to different cohorts is a key metric in deciding its clinical utility.the should report about the demographics distribution of the study population (e.g. ethnic groups, age distribution). Moreover, other data about the patient population would be valuable: for instance, were the patients admitted in emergency room? hospitalized?

Reply: We apologize for this issue, which was raised by Reviewer 1 previously. As mentioned on page 3, paragraph 4, one of the main activities of the Fleury Group is laboratory exams in its 36 laboratories scattered across Brazil. Since it is not a Hospital, we do not have access to information regarding prognostic. Once the patient has performed an exam, there is no further contact. For ethical reasons, the laboratories also do not collect information regarding ethnicity or income. As such, we are unable to perform analysis on these strata. However, age and sex distributions are present in Figure 2 and Table 1 respectively. In our Future Work, delineated on page 16, we wish to implement our approach in a hospital setting which would allow us to collect such information. A disclaimer in the form of a footnote on page 2 formally defines the scope of the data.

Reviewer point P.3.2: Should be more clear on the scope and applicability of their model. Can it be applied to countries other than Brazil

Reply: We believe CBC to be simple and widespread enough to allow our method to be applied to any country other than Brazil. However, we also understand that ethnicity influences CBC results which could affect the model, as well as the machinery employed which may vary. Nevertheless, even though the Brazilian model might not be directly applicable to the whole world, our approach is general enough that one could replicate it anywhere on Earth if data from a specific region is collected. A disclaimer was added in our analysis reflecting this issue on page 15, lines 387-393.

Reviewer point P.3.3: Interestingly, significant changes in CBC parameters were observed

on the day of the first RT-PCR positive. Can this phenomenon be observed in all data sets? Author should discuss the possible reasons.

Reply: We added a passage on page 7, lines 204-210 with our hypothesis behind these patterns and why such behavior appears to be unique to COVID-19, we thank the reviewer for the commentary and hope that our explanation makes this fact more clear.

Reviewer point P.3.4: Abnormal white blood cell values may be a sign of various infections, and it is not uncommon for patients with new coronavirus to have other respiratory virus, bacterial, and fungal infections. You should consider the possible impact of co-infection on the applicability of the model.

Reply: The reviewer is correct and this is indeed a point of attention. Unfortunately, we do not have data on co-infections and as such we cannot provide an insight on this specific issue. We appended a passage on our Future Work, page 16, lines 406-407, regarding collecting this data. We also included a disclaimer concerning our results on page 15, lines 387-390.

Reviewer point P.3.5: The performance of the First-wave model deteriorates over time. Can the author give a more detailed explanation? Please discuss. The experiment emphasized the importance of retraining the models so that they can explain the virus mutation. Affected the applicable value of the model?

Reply: We added a passage on Section 5 Discussion, page 15 lines 372-380 with some of our hypotheses for model degradation as well as our solution. We believe that this in no way affects the applicability of the model, but rather highlights the importance of properly continuously collecting up-to-date and high-quality medical data. We thank the reviewer for pointing this out.

Reviewer point P.3.6: No ethical information being given.

Reply: The reviewer is correct in that this information is missing. This work was evaluated by the Research Ethics Committee (CEP) of Grupo Fleury (CAAE: 33790820.3.0000.5474), qualified by the National Research Ethics Committee (CONEP) of the National Health Council of Brazil. We added a passage on page 2 and 3, lines 120-125 with this information.

Reviewer point P.3.7: According to the IJMEDI checklist for assessment of medical AI, there is a lot of unreachable information in your data set, you should discuss the limitations

Reply: We understand that some information regarding demography is not present as we do not have access to this data and we apologize, but we also believe that our work conforms to most of the points raised in the IJMEDI checklist. Following is a brief discussion regarding each specific topic:

Problem Understanding:

1. We state that we look at patients older than 18 years old with CBC results in a 24h window of an RT-PCR result. (Page 2, paragraph 3)

2. The data was collected before the start of this project, so it is in essence a retrospective study. (page 2, paragraph 2 and page 3, paragraph 5)
3. We employ only laboratory data as explicitly stated on the footnote of page 2.
4. Data were collected from patients attending one of the 36 Fleury laboratories scattered over the country. (page 4, paragraph 1)
5. The task is diagnosis detection as described in the title and multiple instances across the manuscript
6. The machinery and standards employed are described (page 4, paragraph 1 and 2)

Data Understanding:

7. Age demography is described in Figure 2, page 8. Gender demography is described in Table 1, page 5. As addressed in the response, since one of the main activities of the Fleury Group are laboratory exams, and since Fleury is not a Hospital, we do not have access to the ethnic group, comorbidities, or socioeconomic status, some of these mainly for ethical reasons.
8. The gold standard would be the RT-PCR themselves taking into account the safe labeling approaches described on page 6, paragraph 2.
9. Variables are described in Table 2, page 6.

Data Preparation:

10. No outlier detection was applied, all the patients that fit the inclusion-exclusion criteria were evaluated.
11. We did not work with missing values.
12. Feature pre-processing steps are described on page 4, paragraph 3 to page 5, paragraph 2.
13. Data imbalance can be visualized in Table 1, page 5. Class unbalances on the training set are handled mostly by downsampling the COVID-19(-) class. We added an explicit passage with this statement in Table 1 description.

Modeling

14. The model task is reported as a binary classification task, albeit only at the Conclusion. We also included this definition earlier in the manuscript, on page 2, line 33.
15. The model outputs a probability of infection as stated multiple times in the manuscript. Figure 9 on page 13 illustrates the full probability distribution for one of the test sets.
16. Model architecture and hyperparameters are discussed on page 8, paragraph 1.

Validation

17. The validation splits are described on page 5, Table 1. As described on page 4, paragraph 5, our pre-processing step does not use the information present in the validation sets, so no data leakage is present. We also perform 5-fold cross-validation, as expressed on page 8, paragraph 3, and page 9, paragraph 2.
18. Hyperparameter optimization is briefly discussed on page 8, paragraph 3.
19. No calibration was performed.
20. We split validation sets according to timeframes (page 2, paragraph 1) and also create synthetic sets to validate future hypotheses concerning either endemic or large scale circulation of COVID-19 (page 12, paragraph 3)
21. External validation lies in our future work on page 16, paragraph 2.
22. The area under the receiver operating characteristic (auROC), sensitivity, and specificity are reported.
23. We present the model performance deterioration as the validation timeframe distances from the training timeframe, likely due to mutations and changes in protocols.

Deployment

24. In the Conclusion (page 16), we state that the target user is health professionals, suggesting our model to be employed to aid in medical decision making.
25. No baselines for this specific dataset were applied, but we present the results of related work in similar datasets.
26. Model interpretability is provided through the SHAP plots.
27. We discuss bias extensively regarding the presence (or absence) of confounding diseases in COVID-19 related studies and explicitly address this in our work. As we do not have access to demography data, we cannot address these other types of biases.
28. No carbon footprint analyses were made. To the authors' knowledge, there is no such work regarding carbon footprint and LightGBM models, as it is more relevant concerning large neural networks.
29. Data and code availability statements are provided on page 17, paragraph 1.
30. Daily practice adoption lies in our future work on page 16, paragraph 2.

REVIEWERS' COMMENTS:

Reviewer #1 (Remarks to the Author):

I thank the authors for addressing my remarks adequately. Still some problems remain but they are due to the dataset. For me, the manuscript can be accepted.

Reviewer #3 (Remarks to the Author):

all comments were addressed

Belo Horizonte, Brazil, February 21, 2022

We would like to thank once again the editor and reviewers for their interest in our work as well as their comments and suggestions, which we considered to improve our manuscript. In this document, we answer and discuss all the points raised by the reviewers.

Reviewer 1

I thank the authors for addressing my remarks adequately. Still some problems remain but they are due to the dataset. For me, the manuscript can be accepted.

Reply: We are pleased that our work met the Reviewer's standards. We are working on improving the dataset for future work and would like to thank you for the helpful insights provided so far, which are sure to improve our research.

Reviewer 3

all comments were addressed

Reply: We thank the reviewer for all the commentaries and suggestions, which helped us improve our work.